# QUALITY DIVERSITY IMITATION LEARNING

## ABSTRACT

Imitation learning (IL) has shown great potential in various applications, such as robot control. However, traditional IL methods are usually designed to learn only one specific type of behavior since demonstrations typically correspond to a single expert. In this work, we introduce the first generic framework for Quality Diversity Imitation Learning (QD-IL), which enables the agent to learn a broad range of skills from limited demonstrations. Our framework integrates the principles of quality diversity with adversarial imitation learning (AIL) methods, and can potentially improve any inverse reinforcement learning (IRL) method. Empirically, our framework significantly improves the QD performance of GAIL and VAIL on the challenging continuous control tasks derived from Mujoco environments. Moreover, our method even achieves 2x expert performance in the Humanoid environment.

## 1 INTRODUCTION

Imitation learning (IL) enables intelligent systems to quickly learn complex tasks by learning from demonstrations, which is particularly useful when manually designing a reward function is difficult. IL has been applied to many real-world scenarios such as autonomous driving (Bojarski, 2016), robotic manipulation (Zhu et al., 2018), surgical skill learning (Gao et al., 2014), and drone control (Ross et al., 2013). The concept of IL relies on the idea that experts can showcase desired behaviors, when they are unable to directly code them into a pre-defined program. This makes IL applicable to any system requiring autonomous behavior that mirrors expertise (Zare et al., 2024).

However, traditional IL methods tend to replicate only the specific strategies demonstrated by the expert. If the expert demonstrations cover a narrow range of scenarios, the model may struggle when faced with new or unseen situations. Additionally, IL faces challenges in stochastic environments where outcomes are uncertain or highly variable. Since the expert's actions may not capture all possible states or contingencies, IL often struggles to learn an optimal strategy for every scenario (Zare et al., 2024). These limitations are further exacerbated when the demonstration data is limited, as the IL algorithm will only learn specific expert behavior patterns. Hence, traditional IL methods are significantly constrained due to the lack of ability to learn diverse behavior patterns to adapt to stochastic and dynamic environments.

On the other hand, the Quality Diversity (QD) algorithm is designed to find diverse (defined by measure $m$) solutions to optimization problems while maximizing each solution's fitness value (fitness refers to the problem's objective) (Pugh et al., 2016). For instance, QD algorithms can generate diverse human faces resembling "Elon Musk" with various features, such as different eye colors (Fontaine & Nikolaidis, 2021). In robot control, the QD algorithm excels at training policies with diverse behaviors. This enhances the agent's robustness in handling stochastic situations (Tjanaka et al., 2022). For example, if an agent's leg is damaged, it can adapt by switching to a policy that uses the other undamaged leg to hop forward. Different ways of moving forward represent diverse behavior patterns (Fontaine & Nikolaidis, 2021). Traditional QD algorithms often use evolutionary strategies (ES). They have been successful in exploring solution space but suffer from lower fitness due to the large solution spaces, especially when the solution is parameterized by neural networks (Hansen, 2006; Salimans et al., 2017). Recent works combining ES with gradient approximations in differentiable QD (DQD) have significantly improved the ability to discover high-performing and diverse solutions (Fontaine & Nikolaidis, 2021). Naturally, one valuable question is raised: can we design a novel IL framework that can combine the respective strengths of traditional IL and QD algorithms, enabling the agent to learn a broad set of high-performing skills from limited

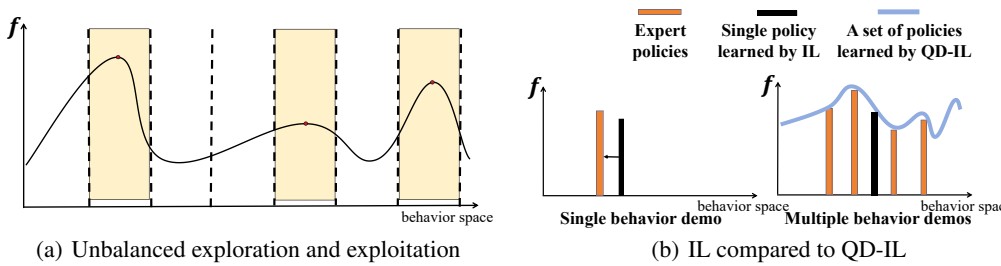

(a) Unbalanced exploration and exploitation    (b) IL compared to QD-IL

Figure 1: (a) The dashed lines divide the policy space into regions constrained by different measures. While PPGA stores high-performing policies for each behavioral region it explores, it overemphasises particular regions of the behavior space (see yellow bars). Introducing the measure bonus helps to improve this exploration process, encouraging exploration in other behavioral regions. (b) The left figure shows traditional IL, where the agent learns a single policy mimicking the expert. In contrast, QD-IL learns from multiple diverse expert policies, such that many behaviors are considered high-performing, resulting in a set of policies, represented by the curve. The orange bar means the expert policy and $f$ means fitness (cumulative reward).

demonstrations? We call such IL framework as Quality Diversity Imitation Learning (QD-IL). Based on our extensive investigations, we found that there is no existing QD-IL work. To mitigate this gap, we first identify the two key challenges of QD-IL as follows:

1) **Unbalanced exploration and exploitation**: From an optimization perspective, we observed that the objective of QD can be framed as solving multiple optimization problems with varying constraints based on measure $m$. Ideally, the policy should explore all regions equally rather than getting stuck in local optima, as depicted in Figure 1(a). However, policy space contains numerous local optima (Dauphin et al., 2014), leading to a lack of behavior-space exploration.

2) **Localized reward**: Traditional Inverse Reinforcement Learning (IRL) methods are inherently formulated based on a single expert policy, as illustrated in the left figure of Figure 1(b). Such a reward design results in a localized re-

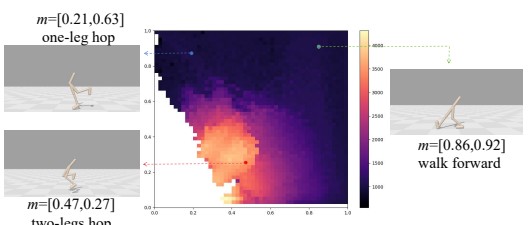

Figure 2: An illustration of the quality-diversity policy archive shows behavior measure $m$, representing the leg ground contact time, where varying $m$ results in diverse behaviors.

ward function, in the sense that it only counts a single behavior as being high-performing. Additionally, the localized reward will further exacerbate the local optima issue mentioned in 1) since we are interested in optimizing a wide range of behaviors, rather than only fitting the expert behavior.

To address these challenges, we introduce two key modifications to generic adversarial IL methods. To improve exploration of new behaviors, we introduce the measure bonus – a reward bonus designed to encourage exploration of new behavior patterns, preventing stagnation at local optima and promoting balanced exploration. To prevent overly localized reward functions, we make two further modifications, namely a) we assume demonstrations are sampled from diverse behaviors from different experts rather than a single expert, as illustrated in the right figure of Figure 1(b); and b) using such diverse demonstrations, we formulate measure conditioning, which enhances the discriminator by incorporating the behavior measure $m$ into its input. The measure $m$ acts as a high-level state abstraction, enabling the generalization of the knowledge from limited demonstrations to unseen states. The measure bonus also promotes the exploration of more diverse state and action pairs. Combined with measure conditioning, this helps reduce the overfitting of the discriminator and addresses the localized reward issue. By combining the measure bonus with measure conditioning, we ensure continuous discovery of new behaviors while generalizing the behavior-level knowledge to unseen situations so that the agent can learn diverse and high-performing policies, as illustrated in Figure 2. To validate our framework, we conducted experiments with limited expert demonstrations across various environments. Notably, our framework is the first generic QD-IL approach, potentially capable of enhancing any IRL method for QD tasks and also opens the possibility of Quality-Diversity Imitation From Observation (QD-IFO) (Liu et al., 2018). It even surpasses expert performance in

terms of both QD-score and coverage in the Walker2d and challenging Humanoid environments. We summarize our contributions as follows:

- We design a measure-based reward bonus to directly encourage behavior-level exploration, which can be integrated into any IRL methods, maximizing the behavior space diversity.
- We propose a novel measure-conditional adversarial IL to generalize expert knowledge to diverse behaviors, which can be applied to most generic IRL algorithms.
- We identify the key challenges of QD-IL. To the best of our knowledge, this paper is the first work to bridge QD algorithms and a broad range of imitation learning methods, addressing the key limitation of traditional IL methods. Our framework provides a generic framework for future QD-IL research and potentially enhances any IL application that requires learning diverse policies.

## 2 BACKGROUND

### 2.1 QUALITY DIVERSITY OPTIMIZATION

Distinct from traditional optimization which aims to find a single solution to maximize the objective, Quality Diversity (QD) optimization aims to find a set of high-quality and diverse solutions in an $n$-dimensional continuous space $\mathbb{R}^n$. Given an objective function $f : \mathbb{R}^n \to \mathbb{R}$ and $k$-dimensional measure function $m : \mathbb{R}^n \to \mathbb{R}^k$, the goal is to find solutions $\theta \in \mathbb{R}^n$ for each local region in the behavior space $B = m(\mathbb{R}^n)$. Two canonical algorithms of QD are MAP-Elites Mouret & Clune (2015a) and Novelty Search with Local Competition Lehman & Stanley (2011), which differ in terms of how they structure the behavior space into an archive of solutions and how local competition and replacement of solutions is performed (see also Cully & Demiris (2018) for an overview of QD algorithm classifications). We focus on grid-based archives as in the MAP-Elites algorithm, which discretize $B$ into $M$ cells, where each cell $i = 1, \ldots, M$ represents a small hypercube $[a_i, b_i]$ within a multi-dimensional grid of the behavioral measure space. A new solution replaces an existing solution in the same cell if it outperforms it and falls within the same hypercube. Formally, the objective is to find a set of solutions $\{\theta_i\}_{i=1}^M$ which maximises $f(\theta_i)$ for each $i = 1, \ldots, M$. Each solution $\theta_i$ corresponds to a cell in $\mathscr{A}$ via its measure $m(\theta_i)$, forming an archive of high-quality and diverse solutions (Chatzilygeroudis et al., 2021; Pugh et al., 2016).

Some traditional Quality Diversity optimization methods integrate Evolution Strategies (ES) with MAP-Elites (Mouret & Clune, 2015b), such as Covariance Matrix Adaptation MAP-Elites (CMA-ME) (Fontaine et al., 2020). CMA-ME uses CMA-ES (Hansen & Ostermeier, 2001) as ES algorithm generating new solutions that are inserted into the archive, and uses MAP-Elites to retains the highest-performing solution in each cell. CMA-ES adapts its sampling distribution based on archive improvements from offspring solutions. However, traditional ES faces low sample efficiency, especially for high-dimensional parameters such as neural networks.

Differentiable Quality Diversity (DQD) improves exploration and fitness by leveraging the gradients of both objective and measure functions. Covariance Matrix Adaptation MAP-Elites via Gradient Arborescence (CMA-MEGA) (Fontaine & Nikolaidis, 2021) optimizes both objective function $f$ and measure functions $m$ using gradients with respect to policy parameters: $\nabla f = \frac{\partial f}{\partial \theta}$ and $\nabla m = \left( \frac{\partial m_1}{\partial \theta}, \ldots, \frac{\partial m_k}{\partial \theta} \right)$. The objective of CMA-MEGA is $g(\theta) = |c_0| f(\theta) + \sum_{j=1}^k c_j m_j(\theta)$, where the coefficients $c_j$ are sampled from a search distribution. CMA-MEGA maintains a search policy $\pi_{\theta_\mu}$ in policy parameter space, corresponding to a specific cell in the archive. CMA-MEGA generates local gradients by combining gradient vectors with coefficient samples from CMA-ES, creating branched policies $\pi_{\theta_1}, \ldots, \pi_{\theta_\lambda}$. These branched policies are ranked based on their archive improvement, which measures how much they improve the QD-score (one QD metric, which will be discussed in the experiment section) of the archive. The ranking guides CMA-ES to update the search distribution, and yields a weighted linear recombination of gradients to step the search policy in the direction of greatest archive improvement. The latest DQD algorithm, Covariance Matrix Adaptation MAP-Annealing via Gradient Arborescence (CMA-MAEGA) (Fontaine & Nikolaidis, 2023), introduces soft archives, which maintain a dynamic threshold $t_e$ for each cell. This threshold is updated by $t_e \leftarrow (1 - \alpha)t_e + \alpha f(\pi_{\theta_i})$ when new policies exceed the cell's threshold, where $\alpha$ balances the time spent on exploring one region before exploring another region. This adaptive mechanism allows more flexible optimization by balancing exploration and exploitation.

## 2.2 QUALITY DIVERSITY REINFORCEMENT LEARNING

The Quality Diversity Reinforcement Learning (QD-RL) problem can be viewed as maximizing $f(\theta) = \mathbb{E}_{\pi_\theta} \left[ \sum_{k=0}^{T-1} \gamma^k r(s_k, a_k) \right]$ with respect to diverse $\theta$ in a policy archive defined by measure $m$ (Cideron et al., 2020). In QD-RL, both the objective and measure are non-differentiable, requiring approximations by DQD approaches. Previous work employs TD3 to approximate gradients and ES for exploration (Nilsson & Cully, 2021; Pierrot et al., 2021), but is constrained to off-policy methods. The state-of-the-art QD-RL algorithm, Proximal Policy Gradient Arborescence (PPGA), employs a vectorized PPO architecture to approximate the gradients of the objective and measure functions (Batra et al., 2023). While the policy gradient can approximate the cumulative reward, the episode-based measure is harder to differentiate. PPGA addresses this by introducing the Markovian Measure Proxy (MMP), a surrogate measure function that correlates strongly with the original measure and allows gradient approximation via policy gradient by treating it as a reward function. PPGA uses $k+1$ parallel environments with distinct reward functions – one for the original reward and $k$ for the surrogate measures. It approximates the gradients of both the objective and the $k$ measure functions by comparing the policy parameters before and after multiple PPO updates. These gradients are then passed to the modified CMA-MAEGA to update the policy archive. We recommend readers to explore prior works in depth (Batra et al., 2023) or refer to Appendix F for further details on PPGA and related QD-RL methodologies.

## 2.3 IMITATION LEARNING

In Imitation learning (IL) (Zare et al., 2024), an agent learns high-performing policies from demonstration data. A traditional approach to solve this challenge is Behavior Cloning (BC), which uses supervised learning to learn the policy from demonstrations, a technique which unfortunately suffers from severe error accumulation (Ross et al., 2011). More recent techniques include inverse reinforcement learning (IRL), where one seeks to learn a reward function from the demonstrations and then use RL to train a policy based on that reward function (Abbeel & Ng, 2004).

Early IRL methods estimate rewards using the principle of maximum entropy (Ziebart et al., 2008; Wulfmeier et al., 2015; Finn et al., 2016). Recent adversarial IL methods treat IRL as a distribution-matching problem. For instance, Generative Adversarial Imitation Learning (GAIL) (Ho & Ermon, 2016) trains a discriminator to differentiate between the state-action distribution of the demonstrations and the state-action distribution induced by the agent's policy, and output a reward to guide policy improvement. Improving on GAIL, Variational Adversarial Imitation Learning (VAIL) (Peng et al., 2018b) applies a variational information bottleneck (VIB) (Alemi et al., 2016) to the discriminator, improving the stability of adversarial learning. Another technique for adversarial IL is Adversarial Inverse Reinforcement Learning (AIRL) (Fu et al., 2017), which learns a robust reward function by training the discriminator via logistic regression to distinguish expert data from policy data.

Recently, one can also observe a variety of techniques for non-adversarial imitation learning. For instance, Primal Wasserstein Imitation Learning (PWIL) (Dadashi et al., 2021) formulates the reward function based on an upper bound of the Wasserstein distance between the expert and agent's state-action distributions, avoiding the instability of adversarial IL methods. Generative Intrinsic Reward-driven Imitation Learning (GIRIL) (Yu et al., 2020) computes rewards offline by pretraining a reward model using a conditional VAE (Sohn et al., 2015), which combines a backward action encoding model with a forward dynamics model. The reward is then derived from the prediction error between the actual next state and its reconstruction. GIRIL has demonstrated superior performance even with limited demonstrations.

Our paper primarily focuses on IRL with limited demonstrations, and we compare our proposed approach to adversarial and non-adversarial techniques as baselines. More details about these baselines are provided in Appendix C. In addition to the IRL setting, we also investigate how our results translate to the related setting of Imitation From Observation (IFO), where only the experts' state sequences, rather than full state-action sequences, are available (Liu et al., 2018).

While we are the first to explore quality diversity imitation learning, related work uses diverse demonstrations to design policies that have the behavioral measure as one of the inputs (Justesen et al., 2020) rather than designing an archive of diverse policies. Subsequent to our work, WQDIL Yu et al. (2024) introduces a technique closely related to ours in terms of recognising behavior-space

Figure 3: MConbo-IRL: Based on episodes sampled from the current search policy, we use the measure conditioned reward model to compute the IRL reward and compare the current archive and the measure of episodes to compute the measure bonus. Then VPPO uses these reward values to approximate gradients for the objective and measures. Then these gradients are used to produce new solutions, update archive, update search distribution, and search policy based on the CMA-MAEGA paradigm.

exploration and measure-conditioning. Three key differences are that WQDIL uses a single-step archive for computing exploration bonus, applies a Wasserstein auto-encoder (WAE) with latent adversarial training, and uses the measure conditioning for learning latent variables with a WAE.

## 3 PROBLEM DEFINITION

**Definition 1** (Quality-Diversity Imitation Learning). *Given expert demonstrations $\mathscr{D} = \{(s_i, a_i)\}_{i=1}^{n}$ and their measures, where $s_i$ and $a_i$ are states and actions, QD-IL aims to learn an archive of diverse policies $\{\pi_{\theta_i}\}_{i=1}^{M}$ that collectively maximizes $f(\theta)$ (e.g., cumulative reward) without access to the true reward. The archive is defined by a k-dimensional measure function $m(\theta)$, representing behavior patterns. After dividing the archive into M cells, the objective of QD-IL is to find M solutions, each occupying one cell, to maximize:*

$$\max_{\{\theta_i\}} \sum_{i=1}^{M} f(\theta_i). \tag{1}$$

## 4 PROPOSED METHOD

In this section, we will introduce our QD-IL framework, which aims to learn a QD-enhanced reward function using the QD-RL algorithm PPGA to learn the policy archive. Specifically, we propose the **measure bonus** to address the challenge of unbalanced exploration and exploitation and **measure conditioning** to address the challenge of localized reward. Figure 3 shows the main components of our framework, PPGA with Measure conditioned and bonus-driven Inverse Reinforcement Learning (PPGA with MConbo-IRL). We provide the pseudo-code of our framework in Appendix A.

### 4.1 MEASURE BONUS

The objective of QD-RL optimization in PPGA is: $g(\theta) = |c_0| f(\theta) + \sum_{j=1}^{k} c_j m_j(\theta)$, where dynamic coefficients $c_i$ balance maximizing cumulative reward $f(\theta)$ and achieving diverse measures $m(\theta)$. However, we observed that the fitness term $f$ heavily influences PPGA's search policy update direction, as archive improvement is primarily driven by $f$. PPGA frequently becomes stuck in local regions, generating overlapping solutions with only marginal improvements in the archive due to limited exploration. Therefore, it will explore less in other areas, as illustrated in Figure 1(a). Additionally, a key challenge in QD-IL is the conflict between imitation learning and diversity. Limited and

monotone expert demonstrations lead to highly localized and sometimes misleading reward functions, further exacerbating the problem by restricting search policy updates. Hence, we aim to encourage the search policy to find new behavior patterns (i.e., the empty area in the policy archive).

**Lemma 1.** *Suppose the reward function of one MDP is given by $r(s_t^i, a_t^i) = \mathbb{I}(m_i \in \mathscr{A}_e)$, where $s_t^i$ and $a_t^i$ represent the state and action at time step $t$ of episode $i$, $\mathscr{A}_e$ means the empty area of archive $\mathscr{A}$ and $\mathbb{I}(m_i \in \mathscr{A}_e)$ is indicator function indicating whether the measure of $i - th$ episode falls into $\mathscr{A}_e$. Then if one iteration of PPO successfully increases the objective value, the following inequalities hold:*

$$(1): P(\pi_{\theta_{new}}|m \in \mathscr{A}_e) \geq P(\pi_{\theta_{old}}|m \in \mathscr{A}_e) \quad and \quad (2): P(m \in \mathscr{A}_e|\pi_{\theta_{new}}) \geq P(m \in \mathscr{A}_e|\pi_{\theta_{old}}),$$

*where $P(m \in \mathscr{A}|\pi_{\theta})$ means the probability of the event that the measure of one episode belongs to the unoccupied area $\mathscr{A}_e$, given this episode is generated by policy $\pi_{\theta}$, and $P(\pi_{\theta}|m \in \mathscr{A}_e)$ means the probability that the policy, which generates the episodes that occupied $\mathscr{A}_e$, is exactly $\pi_{\theta}$.*

Lemma 1 demonstrates that using the indicator function $\mathbb{I}(m_i \in \mathscr{A}_e)$ as the reward function in the standard PPO objective steadily increases the probability that the policy generates episodes with new behavior patterns. We found this approach synergizes effectively with CMA-MEGA, encouraging the search policy to explore diverse behaviors. For the proof of Lemma 1 and a more detailed explanation of the synergy with CMA-MEGA, please refer to Appendix G.

However, we observed that using an indicator function results in binary rewards, which might be sparse and unstable. Moreover, we aim to control the weight of the measure bonus. Hence, we adopt a linear function of indicator for our Measure Bonus:

$$r_{diversity}(s_t^i, a_t^i, m_i) = p + q\mathbb{I}(m_i \in \mathscr{A}_e), \tag{2}$$

where $s_t^i$ and $a_t^i$ represents the state and action at time step $t$ of episode $i$, $\mathscr{A}_e$ means the empty area of current archive and $m_i$ is the measure of episode $i$. The hyperparameter $q$ controls the weight of the measure bonus and the term $p$ encourages the agent for staying in the episode, thereby facilitating the search for diverse behaviors. The trade-off between $p$ and $q$ represents how to emphasise staying in the episode versus getting the measure bonus as frequently as possible. Measure Bonus is a type of *episode reward* (Sutton, 2018), which is calculated at the end of each episode. The Measure Bonus adaptively balances exploration and exploitation. Once a region in the archive has been sufficiently explored, the bonus of this region decreases, allowing the focus to shift more towards exploitation.

## 4.2 MCONBO-IRL

Measure bonus improves policy diversity but doesn't guarantee the performance of diverse policies. To address this, we introduce measure conditioning, which can potentially be integrated into most IRL methods. We demonstrate this using two popular IRL methods, GAIL and VAIL.

### 4.2.1 MCONBO-GAIL

The GAIL discriminator receives a state-action pair $(s, a)$ and outputs how closely the agent's behavior resembles that of the expert, serving as a reward function. However, GAIL tends to overfit specific behaviors with limited demonstrations. In large state spaces, the discriminator struggles to generalize to unseen states (Kostrikov et al., 2018). This results in localized and sparse rewards, hindering quality diversity. Therefore, the core question in QD-IL is how to generalize knowledge from limited demonstrations to the entire policy archive while avoiding localized rewards.

To address this, we use the **Markovian Measure Proxy** (Batra et al., 2023). It decomposes trajectory-based measures into individual steps: $m_i(\theta) = \frac{1}{T}\sum_{t=0}^{T}\delta_i(s_t)$. This makes the measure state-dependent and Markovian. We make the key observation that the single-step measure $\delta_i(s_t)$ abstracts higher-level task features such as ground contact in locomotion, while filtering out lower-level state details (e.g., joint angles and velocities). This provides a more general representation, enabling better generalization across the policy archive. By simply incorporating $\delta_i(s_t)$ as an additional input to the GAIL discriminator, we propose **Measure-Conditional-GAIL** with the following modified objective:

$$\max_{\pi}\min_{D_{\psi}}\mathbb{E}_{(s,a)\sim\mathscr{D}}[-\log D_{\psi}(s, a, \delta(s))] + \mathbb{E}_{(s,a)\sim\pi}[-\log(1 - D_{\psi}(s, a, \delta(s)))]. \tag{3}$$

This approach encourages the discriminator to generalize by focusing on higher-level state descriptors $\delta(s)$, capturing essential task-relevant features. It enables the agent to learn high-performing policies from limited demonstrations, improving generalization to unseen states. The discriminator serves as the basis for the reward, therefore with measure-conditioning the agent will get a high reward for actions that mimic the experts, and particularly so when the expert is close in state-measure, which is an abstraction of the state. Due to it being an abstraction of the state, there is less risk of overfitting on the specific demonstration trajectories since many trajectories may map to the same state. Consequently, many behaviors can be counted as high-performing. Considering the above, the measure-conditioning will help us to achieve high quality and diversity.

We then formulate the total reward function computed by **MConbo-GAIL** as follows:

$$r(s_t^i, a_t^i, m_i) = -\log\left(1 - D_\psi(s_t^i, a_t^i, \delta(s_t^i))\right) + r_{diversity}(s_t^i, a_t^i, m_i). \qquad (4)$$

### 4.2.2 MCONBO-VAIL

To extend our framework to other IRL algorithms, we begin with another generic IL method - Variational Adversarial Imitation Learning (VAIL). To facilitate behavior exploration and knowledge generalization, we slightly modify the VAIL's objective for training discriminator as follows:

$$\min_{D_\psi, E'} \max_{\beta \geq 0} \mathbb{E}_{(s,a)\sim\mathscr{D}}\left[\mathbb{E}_{z\sim E'(z|s,a,\delta(s))}\left[\log(-D_\psi(z))\right]\right] + \mathbb{E}_{(s,a)\sim\pi}\left[\mathbb{E}_{z\sim E'(z|s,a,\delta(s))}\left[-\log(1 - D_\psi(z))\right]\right]$$
$$+ \beta\mathbb{E}_{s\sim\tilde{\pi}}\left[d_{KL}(E'(z|s,a,\delta(s))||p(z)) - I_c\right], \qquad (5)$$

where $\delta(s)$ is the measure proxy function of state $s$, $\tilde{\pi}$ means the mixture of expert policy and agent policy, and $E'$ means latent variable encoder. By simply adding $\delta(s)$ as a new input to the VDB encoder of VAIL, we integrate measure information into the latent variable $z$. This helps improve the generalization ability to diverse behaviors. The reward function for MConbo-VAIL is given by:

$$r(s_t^i, a_t^i, m_i) = -\log\left(1 - D_\psi(\boldsymbol{\mu}_{E'}(s_t^i, a_t^i, \delta(s_t^i)))\right) + r_{diversity}(s_t^i, a_t^i, m_i), \qquad (6)$$

where $\boldsymbol{\mu}_{E'}(s_t^i, a_t^i, \delta(s_t^i))$ represents the mean of encoded latent variable distribution.

## 5 EXPERIMENTS

### 5.1 EXPERIMENT SETUP

We evaluate our framework on three popular Mujoco (Todorov et al., 2012) environments: Halfcheetah, Humanoid, and Walker2d. The goal in each task is to maximize forward progress and robot stability while minimizing energy consumption. Our experiments are based on the PPGA implementation using the Brax simulator (Freeman et al., 2021), enhanced with QDax wrappers for measure calculation (Lim et al., 2022). We leverage pyribs (Tjanaka et al., 2023) and CleanRL's PPO (Huang et al., 2020) for implementing the PPGA algorithm. The observation space sizes for these environments are 17, 18, and 227, with corresponding action space sizes of 6, 6, and 17. The measure function is a vector where each dimension indicates the proportion of time a leg touches the ground. All Experiments are conducted on a system with four A40 48G GPUs, an AMD EPYC 7543P 32-core CPU, and a Linux OS, and each experiment takes roughly two days.

### 5.2 DEMONSTRATIONS

We use a policy archive obtained by PPGA to generate expert demonstrations. To follow a real-world scenario with limited demonstrations, we first sample the top 500 high-performance elites from the archive as a candidate pool. Then from this pool, we select a few demonstrations such that they are as diverse as possible. This process results in 4 diverse demonstrations (episodes) per environment. Appendix B provides the statistical properties, and Figure 4 visualizes the selected demonstrations.

### 5.3 OVERALL PERFORMANCE

To validate the effectiveness of our approach as a generic QD-IL framework, we use the recent state-of-the-art PPGA technique with true reward function as the QD-RL baseline. The PPGA

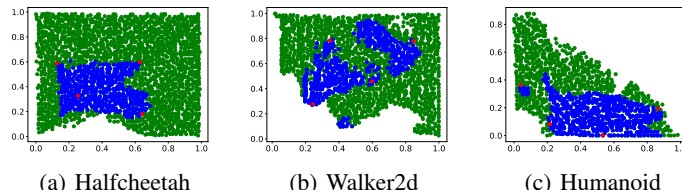

Figure 4: Visualization of the behavior space. Green indicates the full expert behavior space, blue indicates the selected top-500 elites, and red indicates the demonstrators. The x axis is the proportion of time Leg 1 touches the ground and the y axis is the proportion of time Leg 2 touches the ground.

algorithm is then used as the base-learner for our QD-IL imitation learners, by replacing the true reward function with the reward function designed from imitation learning. In addition to using our own MConbo algorithm, we also include the following widely-used and state-of-the-art IL methods as baselines: 1) Traditional IRL: Max-Entropy, 2) Online reward methods: GAIL, VAIL, and AIRL, and 3) Data-driven methods: GIRIL and PWIL. Each baseline learns a reward function, which is then used to train standard PPGA under identical settings for all baselines. Hyperparameter details are provided in Appendix D. All the experiments are averaged with three random seeds, with the exception of PPGA, where we simply report the results for the one seed which was used to generate the demonstrations.

We evaluate using four common QD-RL metrics: 1) **QD-Score**, the sum of scores of all nonempty cells in the archive. QD-score is the most important metric in QD-IL as it aligns with the objective of QD-IL as in equation (1); 2) **Coverage**, the percentage of nonempty cells, indicating the algorithm's ability to discover diverse behaviors; 3) **Best Reward**, the highest score found by the algorithm; and 4) **Average Reward**, the mean score of all nonempty cells, reflecting the ability to discover both diverse and high-performing policies. We use the true reward functions to calculate these metrics.

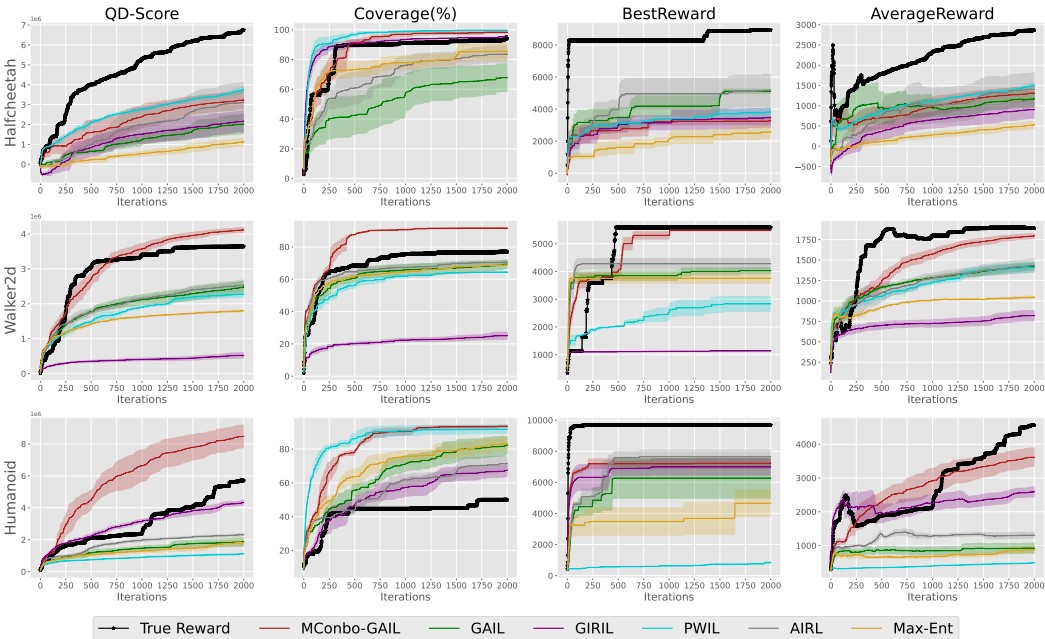

Figure 5: Four QD-metrics for MConbo-GAIL compared to GAIL, PPGA with true reward and other baselines. The line represents the mean while the shaded area represents the standard deviation across three random seeds.

Figure 5 compares the training curves across four metrics for MConbo-GAIL, generic GAIL, the expert (PPGA with true reward function), and other baselines. MConbo-GAIL significantly outperforms the expert in the most challenging Humanoid environment (Batra et al., 2023) and slightly exceeds the expert in the Walker2d environment in terms of QD-Score. In Halfcheetah, MConbo-GAIL improves

the QD performance of generic GAIL and significantly outperforms most baselines across all four metrics. Notably, MConbo-GAIL achieves nearly 100% coverage across all environments, especially notable in Humanoid where the PPGA expert explored less than 50% of the cells. This success is attributed to the synergy between the measure bonus and CMA-MEGA (please refer to Appendix G for detailed explanation), which consistently directs the search policy towards unexplored areas in the behavior space.

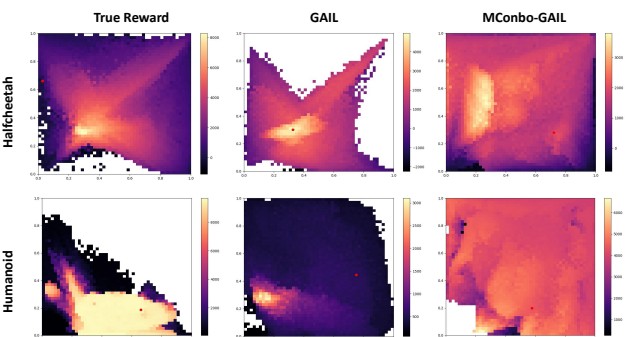

Figure 6: Visualization of well-trained policy archive by True Reward, GAIL and MConbo-GAIL on Humanoid and Halfcheetah, where the color of each cell represents the cumulative reward of best performing policy in this cell.

However, due to the inaccessibility of the true reward function and the limited number of demonstrations, it is challenging to match expert performance in terms of Best Reward and Average Reward. Specifically, these metrics are evaluated using the true reward function, but IL-based reward functions are inherently biased. Despite the biased reward function, MConbo-GAIL achieves near-expert performance in Average Reward for Walker2d and Humanoid. Meanwhile, MConbo-GAIL significantly outperforming GAIL and other baselines in Humanoid and Walker2d for Average Reward. Additionally, it's important to note that GAIL's Average Reward in the Humanoid environment is extremely poor, in stark contrast to MConbo-GAIL's high performance. This can be attributed to the design of Measure-Conditional GAIL, which enables the agent to transfer higher-level knowledge from expert demonstrations to the broader behavior space.

Figure 6 visualizes the policy archives for PPGA expert, GAIL, and MConbo-GAIL in Humanoid and Halfcheetah. The archive produced by MConbo-GAIL shows smoother performance (with lower variance across cells) and covers a larger area, highlighting the importance of measure-space exploration and MConbo-GAIL's effectiveness in generalizing high-level knowledge from limited demonstrations to unseen behavior patterns.

Additionally, to demonstrate the potential of our method to enhance any IRL approach in the QD-IL context, we apply MConbo to the generic VAIL framework. We separately compare MConbo-VAIL with standard VAIL and other baselines, as shown in Figure 8 of Appendix A. Similar conclusions can be drawn: MConbo-VAIL significantly improves VAIL in the QD context and even outperforms the expert in the Walker2d and Humanoid environments.

Table 1: Four QD-metrics of different algorithms across three environments, where cov, Best, Avg refers to Coverage, Best Reward and Average Reward respectively.

| | Halfcheetah | | | | Walker2d | | | | Humanoid | | | |
|---|---|---|---|---|---|---|---|---|---|---|---|---|
| | QD-Score | Cov(%) | Best | Avg | QD-Score | Cov(%) | Best | Avg | QD-Score | Cov(%) | Best | Avg |
| True Reward | $6.75 \times 10^6$ | 94.08 | 8,942 | 2,871 | $3.64 \times 10^6$ | 77.04 | 5,588 | 1,891 | $5.71 \times 10^6$ | 49.96 | 9,691 | 4,570 |
| **MConbo-GAIL** | $3.24 \times 10^6$ | 98.32 | 3,291 | 1,313 | $\mathbf{4.12 \times 10^6}$ | 91.69 | 5,491 | 1,796 | $8.47 \times 10^6$ | 93.47 | 7,228 | 3,618 |
| GAIL | $2.02 \times 10^6$ | 67.83 | 5,115 | 1,167 | $2.48 \times 10^6$ | 69.29 | 4,031 | 1,429 | $1.86 \times 10^6$ | 82.36 | 6,278 | 924 |
| **MConbo-VAIL** | $\mathbf{4.41 \times 10^6}$ | 92.63 | 5,018 | 1,940 | $3.68 \times 10^6$ | 90.60 | 4,051 | 1,626 | $\mathbf{8.91 \times 10^6}$ | 91.52 | 6,505 | 3,899 |
| VAIL | $4.00 \times 10^6$ | 92.77 | 5,167 | 1,724 | $2.40 \times 10^6$ | 71.40 | 3,570 | 1,343 | $5.10 \times 10^6$ | 65.61 | 7,056 | 3,095 |
| GIRIL | $2.17 \times 10^6$ | 95.96 | 3,466 | 909 | $0.52 \times 10^6$ | 25.08 | 1,139 | 821 | $4.33 \times 10^6$ | 67.40 | 6,992 | 2,590 |
| PWIL | $3.75 \times 10^6$ | 99.68 | 3,814 | 1,506 | $2.27 \times 10^6$ | 64.45 | 2,835 | 1,410 | $1.13 \times 10^6$ | 91.73 | 841 | 492 |
| AIRL | $3.11 \times 10^6$ | 83.57 | 5,183 | 1,410 | $2.53 \times 10^6$ | 70.53 | 4,280 | 1,437 | $2.31 \times 10^6$ | 71.47 | 7,661 | 1,308 |
| Max-Ent | $1.12 \times 10^6$ | 85.48 | 2,594 | 525 | $1.80 \times 10^6$ | 68.83 | 3,756 | 1,046 | $1.82 \times 10^6$ | 83.27 | 4,658 | 882 |

Table 1 summarizes the quantitative results of our methods (MConbo-GAIL and MConbo-VAIL) and baselines in the three tasks. MConbo improves boths GAIL and VAIL, thus we believe that our

framework can potentially improve any inverse reinforcement learning algorithm in QD context. We also conducted some experiments for our framework to improve the QD performance of Imitation From Observation (IFO), which is a popular IL branch. Table 2 shows a brief summary of the results. We opened the possibility for quality-diversity imitation-from-observation (QD-IFO). Please refer to Appendix I for detailed analysis.

Table 2: Comparison of four QD-metrics of MConbo-GAIL without expert action (MConbo-GAIL-Obs) and MConbo-GAIL, across three environments. There are only marginal performance losses when expert action is unavailable.

| | Halfcheetah | | | | Walker2d | | | | Humanoid | | | |
|---|---|---|---|---|---|---|---|---|---|---|---|---|
| | QD-Score | Cov(%) | Best | Avg | QD-Score | Cov(%) | Best | Avg | QD-Score | Cov(%) | Best | Avg |
| True Reward | $6.75 \times 10^6$ | 94.08 | 8,942 | 2,871 | $3.64 \times 10^6$ | 77.04 | 5,588 | 1,891 | $5.71 \times 10^6$ | 49.96 | 9,691 | 4,570 |
| MConbo-GAIL-Obs | $3.14 \times 10^6$ | 100.00 | 2,831 | 1,255 | $3.84 \times 10^6$ | 91.02 | 4,940 | 1,689 | $\mathbf{9.28 \times 10^6}$ | 94.02 | 7,759 | 3,936 |
| **MConbo-GAIL** | $\mathbf{3.24 \times 10^6}$ | 98.32 | 3,291 | 1,313 | $\mathbf{4.12 \times 10^6}$ | 91.69 | 5,491 | 1,796 | $8.47 \times 10^6$ | 93.47 | 7,228 | 3,618 |

## 5.4 ABLATION STUDY

In this section, we examine the effect of the measure bonus by comparing the performance of MConbo-GAIL with Measure-Conditional-GAIL (without the measure bonus) on Walker2d. The results in Figure 7 show a significant performance drop across all metrics without the measure bonus. This highlights the synergy between the measure bonus and CMA-MEGA. Without the exploration bonus, the algorithm struggles with highly localized rewards and the inherent local optima of policy gradient approach. As a result, the search policy fails to explore new behavior patterns, leading to lower coverage. Furthermore, since the reward function learned by IL is biased and especially with limited demonstrations, PPGA's search policy may miss opportunities to explore rewarding behavior patterns. This results in lower average and best rewards. The measure bonus directly encourages the exploration of new behaviors, addressing this issue. To view a full ablation study, including the effect of both the measure bonus and measure conditioning, please refer to Appendix E.

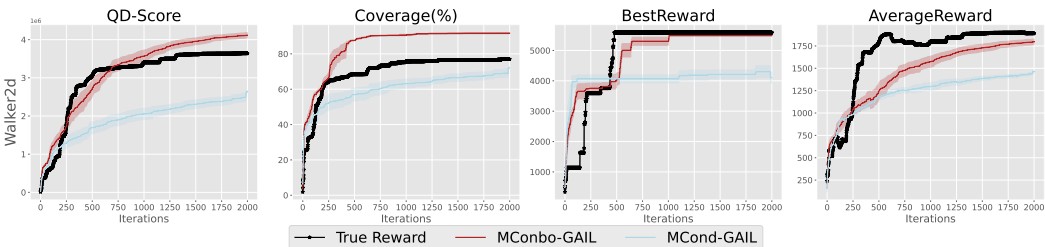

Figure 7: MCond-GAIL means we don't include $r_{diversity}$ into the reward function. The line represents the mean while the shaded area represents the standard deviation across three random seeds.

## 6 CONCLUSION AND FUTURE WORK

In this work, we proposed **MConbo-IRL** which can potentially improve any IRL method in QD task. Additionally, our framework opened the possibility of QD-IFO, providing the first generic QD-IL framework for future research. Our framework follows the paradigm of IRL to learn a QD-enhanced reward function, and use a QD-RL algorithm to optimize policy archive. By encouraging behavior-level exploration and facilitating knowledge generalization from limited expert demonstrations, our framework addresses the key challenges of QD-IL. Extensive experiments show that our framework achieves near-expert or beyond-expert performance, and significantly outperforms baselines.

To establish our framework as a generic QD-IL solution, we focused on improving the two widely used IRL algorithms in this paper to make our framework as simple and effective as possible. However, we believe that our framework has the potential to be compatible with more IRL algorithm backbones. Additionally, exploring the development of new architectures for QD-IL and exciting applications such as behavior adaptation, for instance with context-conditioned policies (Seyed Ghasemipour et al., 2019), remain important avenues for future research. We also discuss the potential limitations of our work in Appendix K.

## 7 REPRODUCIBILITY

We have provided detailed pseudo-code in Appendix A, and a few lists of relevant hyperparameters in Appendix D. In Section 5, we have provided a detailed experiment setup, and a process for generating demonstrations. In supplementary material, we have provided the vedio illustrations of trained diverse behaviors and policy archives.

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

# A   ALGORITHM PSEUDO CODE AND MORE EXPERIMENT RESULTS

Algorithm 1 presents the pseudocode for using MConbo-GAIL as our reward module and PPGA as the QD-RL algorithm. The parts highlighted in red indicate the key distinctions from PPGA. We utilize a reward model to compute the fitness value (reward) for the QD-RL problem, with Algorithm 3 explaining how our reward model functions.

---

**Algorithm 1** PPGA with MConbo-IRL

1: **Input:** Initial policy $\theta_0$, VPPO instance to approximate $\nabla f$, $\nabla m$ and move the search policy, number of QD iterations $N_Q$, number of VPPO iterations to estimate the objective-measure functions and gradients $N_1$, number of VPPO iterations to move the search policy $N_2$, branching population size $\lambda$, and an initial step size for xNES $\sigma_g$. Initial reward model $\mathscr{R}$, Expert data $\mathscr{D}$.
2: Initialize the search policy $\theta_\mu = \theta_0$. Initialize NES parameters $\mu, \Sigma = \sigma_g I$
3: **for** iter $\leftarrow$ 1 to $N$ **do**
4:     $f, \nabla f, \mathbf{m}, \nabla \mathbf{m} \leftarrow$ VPPO.compute_jacobian$(\theta_\mu, \mathscr{R}, \mathbf{m}(\cdot), N_1)$        ▷ approx grad using $\mathscr{R}$
5:     $\nabla f \leftarrow$ normalize$(\nabla f)$,   $\nabla \mathbf{m} \leftarrow$ normalize$(\nabla \mathbf{m})$
6:     _ $\leftarrow$ update_archive$(\theta_\mu, f, \mathbf{m})$
7:     **for** $i \leftarrow$ 1 to $\lambda$ **do** // branching solutions
8:         $c \sim \mathscr{N}(\mu, \Sigma)$ // sample gradient coefficients
9:         $\nabla_i \leftarrow c_0 \nabla f + \sum_{j=1}^k c_j \nabla m_j$
10:        $\theta'_i \leftarrow \theta_\mu + \nabla_i$
11:        $f', *, \mathbf{m}', * \leftarrow$ rollout$(\theta'_i, \mathscr{R})$
12:        $\Delta_i \leftarrow$ update_archive$(\theta'_i, f', \mathbf{m}')$       ▷ get archive improvement of each solution.
13:     **end for**
14:     Rank gradient coefficients $\nabla_i$ by archive improvement $\Delta_i$
15:     Adapt xNES parameters $\mu = \mu', \Sigma = \Sigma'$ based on improvement ranking $\Delta_i$
16:     $f'(\theta_\mu) \leftarrow c_{\mu,0} f + \sum_{j=1}^k c_{\mu,j} m_j$, where $c_\mu = \mu'$
17:     $\theta'_\mu \leftarrow$ VPPO.train$(\theta_\mu, f', \mathbf{m}', N_2, \mathscr{R})$       ▷ walk search policy using reward model $\mathscr{R}$
18:     $\mathscr{R}$.update$(\mathscr{D}, \theta'_\mu)$                              ▷ update reward model
19:     **if** there is no change in the archive **then**
20:        Restart xNES with $\mu = 0, \Sigma = \sigma_g I$
21:        Set $\theta_\mu$ to a randomly selected existing cell $\theta_i$ from the archive
22:     **end if**
23: **end for**

---

**Algorithm 2** Update Archive

**Input:** Solution $\theta$ to insert, episodic reward $f$, measures $\mathbf{m} =< m_1, ..., m_k >$, archive $\mathscr{A}$, archive learning rate $\alpha$
$\theta_{inc}, f_{inc} \leftarrow \mathscr{A}[\mathbf{m}]$ if $\mathscr{A}[\mathbf{m}]$ is nonempty else $None, 0$
$\Delta_i = 0$
**if** $f > f_{inc}$ **then**
    insert $\theta$ into cell $\mathscr{A}[\mathbf{m}]$
    $f_{inc} \leftarrow (1 - \alpha) f_{inc} + \alpha f$
    $\Delta_i = f - f_{inc}$
**end if**
**return** $\Delta_i$

---

---

**Algorithm 3** Reward Model $\mathscr{R}$ (using GAIL as the backbone)

---

1: **Initialize:** Discriminator $D_\phi$
2:
3: **Method: Reward Calculation for VPPO.compute_jacobian()**
4:    `def get_episode_reward`(self, episode, current archive $\mathscr{A}$):
5:      $s_1, a_1, \delta(s_1), s_2, a_2, \delta(s_2) \ldots, s_k, a_k, \delta(s_k) \leftarrow episode$
6:      $r_1, r_2, r_3 \ldots, r_k \leftarrow D_\psi([\mathbf{s}, \mathbf{a}, \delta(\mathbf{s})])$                $\triangleright$ GAIL batch reward
7:      $m \leftarrow episode.get\_measure()$
8:      $r_{diversity} \leftarrow p + q\mathbf{I}(m \in \mathscr{A})$            $\triangleright$ calculate measure bonus
9:      **For** $i = 1 \to k$
10:        $r_i \leftarrow r_i + r_{diversity}$                $\triangleright$ calculate total reward
11:      **return** $r_1, r_2, r_3 \ldots, r_k$
12:
13: **Method: Update reward model**
14:    `def update`(self, $\mathscr{D}, \pi_\theta$):
15:      Sample a batch of trajectories $(\mathbf{s}^\pi, \mathbf{a}^\pi, \delta(\mathbf{s}^\pi))$ from $\pi_\theta$
16:      Update discriminator $D_\psi$ by minimizing:

$$\mathscr{L}_D(\psi) = \mathbb{E}_{(\mathbf{s},\mathbf{a})\sim\mathscr{D}}[-\log D_\psi(\mathbf{s}, \mathbf{a}, \delta(\mathbf{s}))] + \mathbb{E}_{(\mathbf{s},\mathbf{a})\sim\pi_\theta}[-\log(1 - D_\psi(\mathbf{s}, \mathbf{a}, \delta(\mathbf{s})))]$$

17:      Repeat until the model converges or the number of epochs is reached
18:      **return** Updated $D_\psi$

---

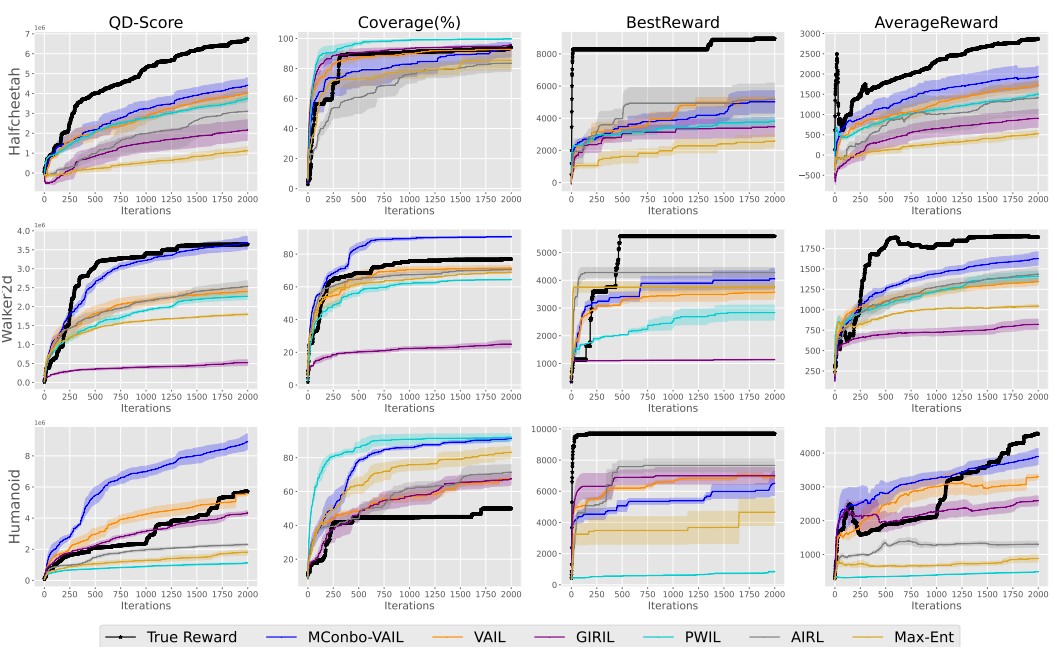

Figure 8: Four QD-metrics for MConbo-VAIL compared to VAIL, PPGA with true reward and other baselines. The line represents the mean while the shaded area represents the standard deviation across three random seeds.

## B   DEMONSTRATION DETAILS

Figure 9 shows the Mujoco environments used in our experiments. Table 3 shows the detailed information of the demonstrations in our experiment.

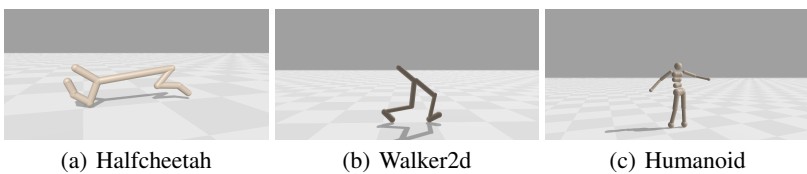

|              | (a) Halfcheetah | (b) Walker2d | (c) Humanoid |

Figure 9: Mujoco Environments.

Table 3: Demonstrations are generated from **top-500** high-performance elites.

| Tasks | Demo number | Attributes | min | max | mean | std |
|-------|-------------|------------|-----|-----|------|-----|
| Halfcheetah | 4 | Length | 1000 | 1000 | 1000.0 | 0.0 |
|  |  | Demonstration Return | 3766.0 | 8405.4 | 5721.3 | 1927.6 |
| Walker2d | 4 | Length | 356.0 | 1000.0 | 625.8 | 254.4 |
|  |  | Demonstration Return | 1147.9 | 3721.8 | 2372.3 | 1123.7 |
| Humanoid | 4 | Length | 1000.0 | 1000.0 | 1000.0 | 0.0 |
|  |  | Demonstration Return | 7806.2 | 9722.6 | 8829.5 | 698.1 |

## C  BASELINE IMITATION LEARNING METHODS

This section summarizes the details for the related IRL methods used as baselines in this paper:

- **GAIL** (Ho & Ermon, 2016). In GAIL, the objective of the discriminator $D_\psi$ is to differentiate between the state-action distribution of expert demonstration $\mathscr{D}$ and the state-action distribution induced by the agent's policy $\pi$:

$$\max_\pi \min_{D_\psi} \mathbb{E}_{(s,a)\sim\mathscr{D}}[-\log D_\psi(s,a)] + \mathbb{E}_{(s,a)\sim\pi}[-\log(1-D_\psi(s,a))]. \tag{7}$$

  The discriminator is trained to maximize the likelihood assigned to states and actions from the target policy while minimizing the likelihood assigned to states and actions from the agent's policy. The discriminator also serves as the agent's reward function, encouraging the policy to visit states that, to the discriminator, appear indistinguishable from the demonstrations. The reward for $\pi$ is then specified by the discriminator $r_t = -\log\big(1 - D_\psi(s,a)\big)$.

- **VAIL** (Peng et al., 2018a) improves GAIL by compressing the information via a variational information bottleneck (VDB). VDB constrains information flow in the discriminator by means of an information bottleneck. By enforcing a constraint on the mutual information between the observations and the discriminator's internal representation, VAIL significantly outperforms GAIL by optimizing the following objective:

$$\min_{D_\psi, E'} \max_{\beta \geq 0} \mathbb{E}_{(s,a)\sim\mathscr{D}}\Big[\mathbb{E}_{z\sim E'(z|s,a)}\big[\log(-D_\psi(z))\big]\Big] + \mathbb{E}_{(s,a)\sim\pi}\Big[\mathbb{E}_{z\sim E'(z|s,a)}\big[-\log(1-D_\psi(z))\big]\Big]$$
$$+ \beta\mathbb{E}_{s\sim\tilde{\pi}}\big[d_{KL}(E'(z|s,a)||p(z)) - I_c\big], \tag{8}$$

  where $\tilde{\pi} = \frac{1}{2}\pi^E + \frac{1}{2}\pi$ represents a mixture of the expert policy and the agent's policy, $E'$ is the encoder for VDB, $\beta$ is the scaling weight, $p(z)$ is the prior distribution of latent variable $z$, and $I_c$ is the information constraint. The reward for $\pi$ is then specified by the discriminator $r_t = -\log\big(1 - D_\psi(\boldsymbol{\mu}_{E'}(s_t, a_t))\big)$.

- **AIRL** (Fu et al., 2017) is an inverse reinforcement learning algorithm based on adversarial learning. AIRL leverages binary logistic regression to train the discriminator to classify expert data and the agent's policy data. The reward $r$ is updated in terms of $r(s,a,s') \leftarrow \log D_\psi(s,a,s') - \log(1 - D_\psi(s,a,s'))$.

- **GIRIL** (Yu et al., 2020). Previous inverse reinforcement learning (IRL) methods usually fail to achieve expert-level performance when learning with limited demonstrations in high-dimensional environments. To address this challenge, Yu et al. (2020) proposed generative intrinsic reward-driven imitation learning (GIRIL) to empower the agent with the demonstrator's intrinsic intention

and better exploration ability. This was achieved by training a novel reward model to generate intrinsic reward signals via a generative model. Specifically, GIRIL leverages a conditional VAE (Sohn et al., 2015) to combine a backward action encoding model and a forward dynamics model into a single generative model. The module is composed of several neural networks, including recognition network $q_\phi(z|s_t, s_{t+1})$, a generative network $p_\varphi(s_{t+1}|z, s_t)$, and prior network $p_\varphi(z|s_t)$. GIRIL refers to the recognition network (i.e. the probabilistic *encoder*) as a backward action encoding model, and the generative network (i.e. the probabilistic *decoder*) as a forward dynamics model. Maximizing the following objective to optimize the module:

$$J(p_\varphi, q_\phi) = \mathbb{E}_{q_\phi(z|s_t, s_{t+1})}[\log p_\varphi(s_{t+1}|z, s_t)] - \mathrm{KL}(q_\phi(z|s_t, s_{t+1}) \| p_\varphi(z|s_t))$$
$$- \alpha d_{KL}(q_\phi(\hat{a}_t|s_t, s_{t+1}) | \pi_E(a_t|s_t)), \tag{9}$$

where $z$ is the latent variable, $\pi_E(a_t|s_t)$ is the expert policy distribution, $\hat{a}_t = \mathrm{Softmax}(z)$ is the transformed latent variable, $\alpha$ is a positive scaling weight. The reward model will be pre-trained on the demonstration data and used for inferring intrinsic rewards for the policy data. The intrinsic reward is calculated as the reconstruction error between $\hat{s}_{t+1}$ and $s_{t+1}$:

$$r_t = \|\hat{s}_{t+1} - s_{t+1}\|_2^2, \tag{10}$$

where $\|\cdot\|_2$ denotes the L2 norm, $\hat{s}_{t+1} = decoder(a_t, s_t)$.

- **PWIL** (Dadashi et al., 2021) introduces a reward function based on an upper bound of the Wasserstein distance between the state-action distributions of the agent ($\pi$) and the expert demonstrations (i.e. the data from $\mathscr{D}$). The Wasserstein distance is defined as:

$$\inf_{\pi \in \Pi} \mathscr{W}_p^p(\hat{\rho}_\pi, \hat{\rho}_e) = \inf_{\pi \in \Pi} \inf_{\omega \in \Omega} \sum_{i=1}^{T} \sum_{j=1}^{N} d((s_i^\pi, a_i^\pi), (s_j^e, a_j^e))^p \omega[i, j], \tag{11}$$

where $\pi$ is the policy, and $\omega[i, j]$ represents the coupling between state-action pairs.

PWIL then defines an upper bound of the Wasserstein distance using a greedy coupling, which provides a suboptimal but efficient way to compute the coupling:

$$\inf_{\pi \in \Pi} \mathscr{W}_1(\hat{\rho}_\pi, \hat{\rho}_e) = \inf_{\pi \in \Pi} \sum_{i=1}^{T} \sum_{j=1}^{N} d((s_i^\pi, a_i^\pi), (s_j^e, a_j^e)) \omega_\pi^*[i, j]$$
$$\leq \inf_{\pi \in \Pi} \sum_{i=1}^{T} \sum_{j=1}^{N} d((s_i^\pi, a_i^\pi), (s_j^e, a_j^e)) \omega_\pi^g[i, j], \tag{12}$$

where $\omega_\pi^g$ represents the greedy coupling.

The greedy coupling $\omega_\pi^g$ is defined recursively for each timestep $i$ as:

$$\omega_\pi^g[i, :] = \arg \min_{\omega[i, :] \in \Omega_i} \sum_{j=1}^{N} d((s_i^\pi, a_i^\pi), (s_j^e, a_j^e)) \omega[i, j], \tag{13}$$

where $\Omega_i$ is a feasible set of couplings constrained by:

$$\Omega_i = \left\{ \omega[i, :] \in \mathbb{R}_+^N \;\Big|\; \sum_{j'=1}^{N} \omega[i, j'] = \frac{1}{T}, \forall k \in [1 : N], \sum_{i'=1}^{i-1} \omega_g[i', k] + \omega[i, k] \leq \frac{1}{N} \right\}. \tag{14}$$

Finally, a reward is derived from the cost $c_\pi^g = \sum_{j=1}^{N} d((s_i^\pi, a_i^\pi), (s_j^e, a_j^e)) \omega_\pi^g[i, j]$ by applying a monotonically decreasing function $f$:

$$r_{i,\pi} = f(c_{i,\pi}^g), \tag{15}$$

where the reward $r_{i,\pi}$ is history-dependent. PWIL avoids the inner minimization problem typically found in adversarial imitation learning approaches, focusing instead on maximizing the derived reward directly.

# D  HYPERPARAMETER SETTING

## D.1  HYPERPARAMETERS FOR PPGA

Table 4 summarizes a list of hyperparameters for PPGA policy updates.

Table 4: List of relevant hyperparameters for PPGA shared across all environments.

| Hyperparameter | Value |
|---|---|
| Actor Network | [128, 128, Action Dim] |
| Critic Network | [256, 256, 1] |
| $N_1$ | 10 |
| $N_2$ | 10 |
| PPO Num Minibatches | 8 |
| PPO Num Epochs | 4 |
| Observation Normalization | True |
| Reward Normalization | True |
| Rollout Length | 128 |
| Grid Size | 50 |
| Env Batch Size | 3,000 |
| Num iterations | 2,000 |

## D.2 HYPERPARAMETERS FOR IL

Table 5 summarizes a list of hyperparameters for AIRL, GAIL, measure-conditioned GAIL, and MConbo-GAIL.

Table 5: List of relevant hyperparameters for AIRL, GAILs shared across all environments.

| Hyperparameter | Value |
|---|---|
| Discriminator | [100, 100, 1] |
| Learning Rate | 3e-4 |
| Discriminator Num Epochs | 1 |

Table 6 summarizes a list of hyperparameters for VAIL, measure-conditioned VAIL, and MConbo-VAIL.

Table 6: List of relevant hyperparameters for VAILs shared across all environments.

| Hyperparameter | Value |
|---|---|
| Discriminator | [100, 100, (1, 50, 50)] |
| Learning Rate | 3e-4 |
| Information Constraint $I_c$ | 0.5 |
| Discriminator Num Epoch | 1 |

Table 7 summarizes a list of hyperparameters for GIRIL.

Table 7: List of relevant hyperparameters for GIRIL shared across all environments.

| Hyperparameter | Value |
|---|---|
| Encoder | [100, 100, Action Dim] |
| Decoder | [100, 100, Observation Dim] |
| Learning Rate | 3e-4 |
| Batch Size | 32 |
| Num Pretrain Epochs | 10,000 |

Table 8 summarizes a list of hyperparameters for MConbo-IRL framework. While our measure bonus function $r_{diversity}(s_t^i, a_t^i, m_i) = p + q\mathbb{I}(m_i \in A_e)$ introduces hyperparameters $p$ and $q$, these were not extensively tuned in our experiments. We used $p = q = 0.5$ across all environments, which provided satisfactory performance. However, the optimal values may vary depending on the specific task and environment characteristics.

Table 8: List of relevant hyperparameters for MConbo shared across all environments.

| Hyperparameter | Value |
|:---:|:---:|
| $p$ | 0.5 |
| $q$ | 0.5 |

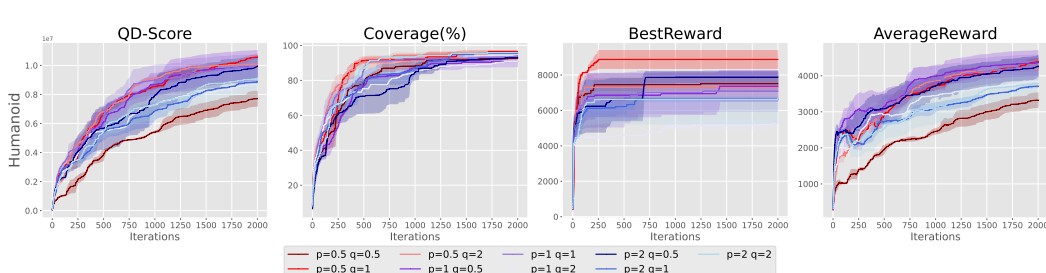

Figure 10: Effect of different $p$ and $q$ values: we observe that $p = 0.5$ and $q = 1$ is the best choice of our experiment, suggesting that our result can be further optimized.

### D.3 HYPERPARAMETER STUDY FOR $p$ AND $q$

We hereby study the effect of different choice of hyperparameter p and q, as illustrated in Figure 10.

## E FULL ABLATION STUDY

To verify the effect of measure conditioning and measure bonus in GAIL, we compare the performance of MCond-GAIL (GAIL with measure conditioning) and MConbo-GAIL (GAIL with measure conditioning and bonus) in all three environments, as illustrated in Figure 7.

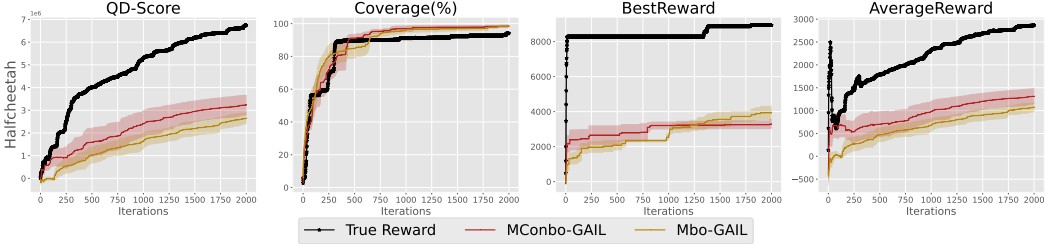

Figure 11: The effect of measure conditioning and measure bonus in GAIL. The line represents the mean while the shaded area represents the standard deviation across three random seeds.

Comparing MCond-GAIL and GAIL, we observe that MCond-GAIL strongly improves on GAIL on all QD metrics in the Halfcheetah environment, while obtaining comparable scores in other environments. Additionally, comparing MConbo-GAIL to MCond-GAIL, we see further improvements, with MConbo-GAIL outperforming MCond-GAIL on coverage and QD-score in all three environments. The table shows that the benefit of MCond-GAIL over GAIL is consistent in the experiments although with varying effect size (2+ pooled std, 1 pooled std, and one very small effect).

## F DETAILS ABOUT PPGA AND RELATED BACKGROUND

To help readers to better understand the background of QD-RL, we begin with Covariance Matrix Adaptation MAP-Elites via a Gradient Arborescence (CMA-MEGA) (Fontaine & Nikolaidis, 2021).

For a general QD-optimization problem, the objective of CMA-MEGA is:

$$g(\boldsymbol{\theta}) = |c_0|f(\boldsymbol{\theta}) + \sum_{j=1}^{k} c_j m_j(\boldsymbol{\theta}), \tag{16}$$

Table 9: The effect of measure conditioning and measure bonus in GAIL in terms of the mean and standard deviation of the metric scores in the final 10 iterations of the runs.

| Tasks | Model | QD-Score | Coverage | BestReward | AverageReward |
|---|---|---|---|---|---|
| Halfcheetah | True Reward | 6,752,624 | 94.08 | 8,942 | 2,871 |
| Halfcheetah | GAIL | 2,022,500±839,063 | 67.83±16.05 | 5,115±218 | 1,167±341 |
| Halfcheetah | MConbo-GAIL | 3,235,423±750,505 | 98.32±1.21 | 3,291±430 | 1,313±291 |
| Halfcheetah | MCond-GAIL | 3,530,990±415,587 | 85.91±13.95 | 5,832±531 | 1,704±406 |
| Humanoid | True Reward | 5,708,191 | 49.96 | 9,691 | 4,570 |
| Humanoid | GAIL | 1,864,664±450,333 | 82.36±9.16 | 6,278±2,245 | 924±250 |
| Humanoid | MConbo-GAIL | 8,470,826±1,235,069 | 93.47±1.37 | 7,228±582 | 3,618±475 |
| Humanoid | MCond-GAIL | 2,194,446±316,492 | 69.07±5.54 | 7,795±397 | 1,266±105 |
| Walker2d | True Reward | 3,641,854 | 77.04 | 5,588 | 1,891 |
| Walker2d | GAIL | 2,483,228±288,096 | 69.29±4.48 | 4,031±187 | 1,429±73 |
| Walker2d | MConbo-GAIL | 4,115,586±119,161 | 91.69±0.58 | 5,491±40 | 1,796±56 |
| Walker2d | MCond-GAIL | 2,501,431±192,553 | 69.23±4.04 | 4,302±348 | 1,444±32 |

In this context, $m_j(\theta)$ represents the $j$-th measure of the solution $\theta$, and $k$ is the dimension of the measure space. The objective function of CMA-MEGA is dynamic because the coefficient for each measure, $c_j$, is updated adaptively to encourage diversity in $m$. For instance, if the algorithm has already found many solutions with high $m_1$ values, it may favor new solutions with low $m_1$ values by making $c_1$ negative, thus minimizing $m_1$. However, the coefficient for the fitness function $f$ will always be positive, as the algorithm always seeks to maximize fitness. This objective function ensures that CMA-MEGA simultaneously maximizes fitness $f$ and encourages diversity across the measures $m$. We update $\theta$ by differentiating objective (16) and use gradient-descend-based optimization approaches, since DQD assumes $f$ and $m$ are differentiable.

Furthermore, the coefficients $c_j$ are sampled from a distribution, which is maintained using Covariance Matrix Adaptation Evolution Strategy (CMA-ES) (Hansen, 2016). Specifically, CMA-ES updates the coefficient distribution by iteratively adapting the mean $\mu$ and covariance matrix $\Sigma$ of the multivariate Gaussian distribution $N(\mu, \Sigma)$, from which the coefficients $c_j$ are sampled. At each iteration, CMA-MEGA ranks the solutions based on their archive improvement (i.e. How much they improve the existing solutions of occupied cell). The top-performing solutions are used to update $\mu$, while $\Sigma$ is adjusted to capture the direction and magnitude of successful steps in the solution space, thereby refining the search distribution over time.

In CMA-MAEGA (Fontaine & Nikolaidis, 2023), the concept of **soft archives** is introduced to improve upon CMA-MEGA. Instead of maintaining the best policy in each cell, the archive employs a dynamic threshold, denoted as $t_e$. This threshold is updated using the following rule whenever a new policy $\pi_{\theta_i}$ surpasses the current threshold of its corresponding cell $e$:

$$t_e \leftarrow (1-\alpha)t_e + \alpha f(\pi_{\theta_i})$$

Here, $\alpha$ is a hyperparameter called the **archive learning rate**, with $0 \leq \alpha \leq 1$. The value of $\alpha$ controls how much time is spent optimizing within a specific region of the archive before moving to explore a new region. Lower values of $\alpha$ result in slower threshold updates, emphasizing exploration in a particular region, while higher values promote quicker transitions to different areas. The concept of soft archives offers several theoretical and practical advantages, as highlighted in previous studies.

PPGA (Batra et al., 2023) is directly built upon CMA-MAEGA. We summarize the key synergies between PPGA and CMA-MAEGA as follows:(1) In reinforcement learning (RL), the objective functions $f$ and $m$ in Equation 16 are not directly differentiable. To address this, PPGA employs **Markovian Measure Proxies (MMP)**, where a single-step proxy $\delta(s_t)$ is treated as the reward function of an MDP. PPGA utilizes $k+1$ parallel PPO instances to approximate the gradients of $f$ and each measure $m$, where $k$ is the number of measures. Specifically, the gradient for each $i$-MDP is computed as the difference between the parameters $\theta_{i,\text{new}}$ after multi-step PPO optimization and the previous parameters $\theta_{i,\text{old}}$. (2) Once the gradients are approximated, the problem is transformed into a standard DQD problem. PPGA then applies a modified version of CMA-MAEGA to perform quality diversity optimization. The key modifications include:

1. **Replacing CMA-ES with xNES for Stability**: To improve stability in noisy reinforcement learning environments, CMA-ES was replaced with Exponential Natural Evolution Strategy (xNES). While CMA-ES struggled with noisy, high-dimensional tasks due to its cumulative step-size adaptation mechanism, xNES provided more stable updates to the search distribution, especially in low-dimensional objective-measure spaces, and maintained search diversity.

2. **Walking the Search Policy with VPPO**: PPGA "walks" the search policy over multiple steps by optimizing a new multi-objective reward function with VPPO (Vectorized Proximal Policy Optimization). This is done by leveraging the mean gradient coefficient vector from xNES, ensuring stable and controlled movement toward greater archive improvement.

## G  SYNERGY OF MEASURE BONUS WITH CMA-MEGA

In our QD-IL framework, we made a key observation that by introducing essential guiding signals into the fitness function $f$, we can effectively encourage exploration at the behavior level.

Firstly, note that in a traditional QD-RL setting, the elite of one cell is a policy $\theta$. However, the performance of this elite is computed by the random episodes produced by the policy $\theta$. Thus, the same policy may produce different episodes which occupy different cells. Hence, the motivation of our method is to improve the probability that the new policy produce episodes occupying the empty area of archive, which is the conclusion of Lemma 1. We first give the proof of Lemma 1:

*Proof.* Proof of (1): The objective of policy optimization is:

$$
h(\theta, \mathscr{A}_e) = \mathbb{E}_{\tau_i \sim \pi_\theta} \left[ \sum_{t=0}^{T} \gamma^t r(s_t^i, a_t^i) \right] = \mathbb{E}_{\tau_i \sim \pi_\theta} \left[ \mathbb{I}(m_i \in \mathscr{A}_e) \cdot \sum_{t=0}^{T} \gamma^t \right] = \frac{1 - \gamma^{T+1}}{1 - \gamma} \mathbb{E}_{\tau_i \sim \pi_\theta} \left[ \mathbb{I}(m_i \in \mathscr{A}_e) \right]
\tag{17}
$$

where $T$ is the episode length (rollout length).

Optimizing $h(\theta_{\text{old}}, \mathscr{A}_e)$ through multiple rounds of PPO will result in $\theta_{\text{new}}$ such that $h(\theta_{\text{new}}, \mathscr{A}_e) > h(\theta_{\text{old}}, \mathscr{A}_e)$, since PPO is assumed to steadily improve the policy, thus increasing the objective.

Therefore, we have:

$$
\mathbb{E}_{\tau_i \sim \pi_{\theta_{\text{new}}}} \left[ \mathbb{I}(m_i \in \mathscr{A}_e) \right] \geq \mathbb{E}_{\tau_i \sim \pi_{\theta_{\text{old}}}} \left[ \mathbb{I}(m_i \in \mathscr{A}_e) \right]
\tag{18}
$$

Since $\mathbb{E}_{\tau_i \sim \pi_\theta} \left[ \mathbb{I}(m_i \in \mathscr{A}_e) \right] = P(m \in \mathscr{A}_e | \pi_\theta)$ where $m_i$ is the measure of episode $\tau_i$, it follows that:

$$
P(m \in \mathscr{A}_e | \pi_{\theta_{\text{new}}}) \geq P(m \in \mathscr{A}_e | \pi_{\theta_{\text{old}}})
$$

Proof of (2): Based on Bayes' rule, we have:

$$
P(\pi_\theta | m \in \mathscr{A}_e) = \frac{P(m \in \mathscr{A}_e | \pi_\theta) P(\pi_\theta)}{P(m \in \mathscr{A}_e)} \propto P(m \in \mathscr{A}_e | \pi_\theta)
$$

Since $P(\pi_\theta)$ and $P(m \in \mathscr{A}_e)$ can be treated as constants when $\theta$ changes (assuming $\theta$ has uniform prior), we have:

$$
P(\pi_{\theta_{\text{new}}} | m \in \mathscr{A}_e) \geq P(\pi_{\theta_{\text{old}}} | m \in \mathscr{A}_e)
$$

Thus, the lemma is proved. $\qquad\square$

It is worthy noting that, 1) while Lemma 1 offers valuable intuition for our approach, our method's practical effectiveness is not solely dependent on the theoretical guarantee of monotonic improvement in PPO. 2) the solution will only be added to the archive during the "update_archive" step in Algorithm 1. However, the scope of Lemma 1 is limited to "VPPO.compute_jacobian" and "VPPO.train". This implies that the episodes generated during the training phase and the gradient-approximating stage will not be inserted into the archive.

If we apply a measure bonus to the original GAIL reward, the objective of CMA-MEGA transforms into:

$$
g(\theta) = |c_0|[f(\theta) + h(\theta, \mathscr{A})] + \sum_{j=1}^{k} c_j m_j(\theta),
\tag{19}
$$

where $h(\theta, \mathscr{A})$ represents the cumulative bonus reward based on the current policy archive $\mathscr{A}$. The gradient of $\theta$ becomes:

$$\nabla_\theta g(\theta) = |c_0|\nabla f(\theta) + |c_0|\nabla_\theta h(\theta, \mathscr{A}) + \sum_{j=1}^{k} c_j \nabla_\theta m_j(\theta).$$

Notably, the fitness function $f(\theta)$ is calculated using the GAIL reward in the QD-IL setting. Lemma 1 shows that the measure bonus leads to a new policy that has a higher probability of producing episodes with measures in the empty regions of the archive. As a result, a higher $c_0$ value will guide the search policy towards unoccupied areas in the archive, leading to significant archive improvements (since occupying a new cell naturally results in larger archive improvements compared to replacing an existing elite in a cell).

Furthermore, based on the properties of CMA-ES, the value of $c_0$ tends to increase temporarily, and the term $|c_0|\nabla_\theta h(\theta, \mathscr{A})$ will dominate, facilitating the search policy's exploration of new behavior patterns. On the other hand, if one area of the archive becomes sufficiently explored, the measure bonus will decrease to a standard level, restoring the relative importance of the fitness term in the objective.

## H  SCALABILITY STUDY

We also explore the scalability of the MConbo framework. The key challenge of QD-IL is learning diverse policies from homogeneous expert demonstrations, so we test MConbo-GAIL's ability to scale with fewer demonstrations, representing more uniform expert behavior. Using the Walker2d environment, we reduce the number of demonstrations to 2 and 1 and compare the performance of MConbo-GAIL and GAIL.

Figure 12 shows the learning curves of MConbo-GAIL, GAIL, and PPGA (true reward), while Figure 13 compares their performance of QD-score and coverage. Notably, the coverage of MConbo-GAIL remains close to 100% despite the decrease in expert demonstration numbers, highlighting the robustness of Measure Bonus to consistently find diverse policies. This robustness is attributed to the synergy between Measure Bonus and CMA-MEGA (Appendix G). On the other hand, fewer demonstrations reduce the quality of expert data, leading to lower QD scores. This is especially true for MConbo-GAIL, which will inherently explore some behavior space regions which is distant from the expert behavior. Hence, learning high-performing policy will be difficult, when the algorithm can't find relevant behavior patterns in expert demonstrations. However, MConbo-GAIL still outperforms GAIL. It can learn diverse and relatively high-performing policies even with just one demonstration, demonstrating its scalability with limited expert data.

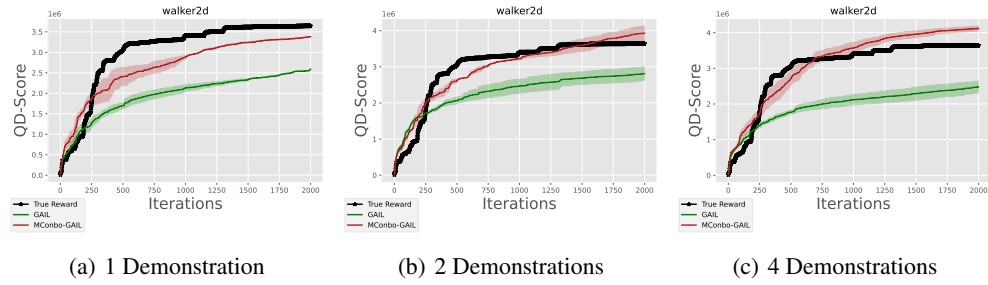

|(a) 1 Demonstration|(b) 2 Demonstrations|(c) 4 Demonstrations|

Figure 12: Scalability Study: we test the effect that limited number of expert demonstrations have on the performance of our MConbo-GAIL model, compared with traditional GAIL. We set the number of demonstrations to 1, 2 and 4. The line represents the mean while the shaded area represents the standard deviation across three random seeds.

Moreover, we compare the training curve between our original setting (4 demos) with 10 demonstrations using the same demonstration sampling method, as illustrated in Figure 14. The result shows more diverse demonstrations will bring higher performance. However, to show the capability of our approach to deal with limited demonstrations, we use only 4 demonstrations in our setting.

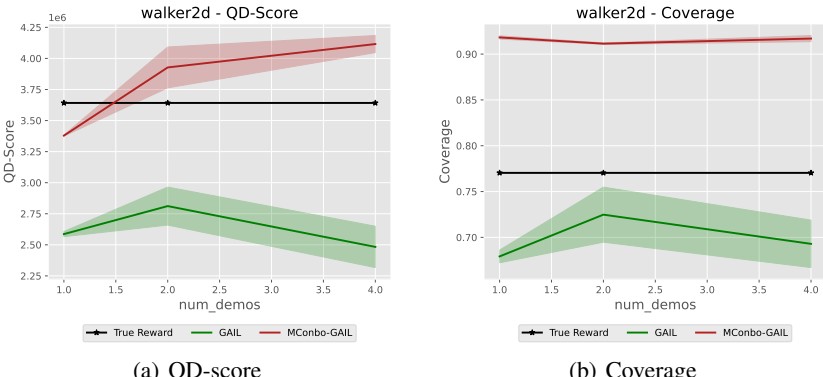

(a) QD-score        (b) Coverage

Figure 13: We compare the performance fluctuation due to decrease of number of demonstrations of MConbo-GAIL and traditional GAIL. The line represents the mean while the shaded area represents the standard deviation across three random seeds.

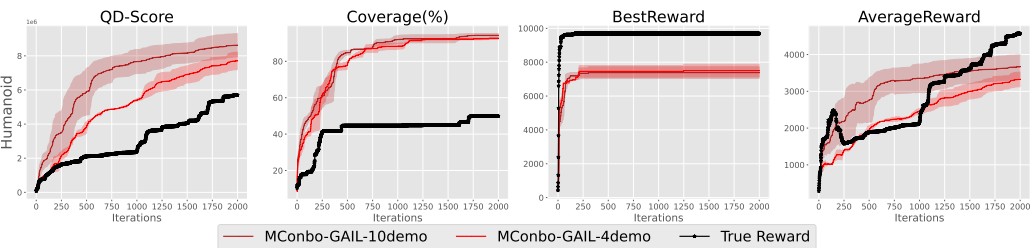

Figure 14: Comparison of performance between 4 demos and 10 demos. The performance of 10 demos is significantly better than 4 demos, suggesting that the more diverse the demonstration, the better the performance of our algorithm.

# I IMPROVE IMITATION FROM OBSERVATION

Imitation from Observation (IFO) is a type of imitation learning where agents learn behaviors by observing state trajectories, without needing access to the actions that generated them. Unlike traditional methods that require both states and actions, IFO mimics behavior solely from state sequences, making it ideal for situations like video demonstrations. This approach aligns more naturally with how humans and animals learn, as we often imitate behaviors by observation without knowing the exact actions involved (e.g., muscle movements) (Zare et al., 2024). IFO is particularly useful in scenarios where action data is unavailable, using techniques like inverse reinforcement learning to infer the underlying policy.

We further observe the potential of our MConbo framework to handle IFO problem, as illustrated in Figure 15. In the setting of IFO, we modify the objective of MConbo-GAIL as:

$$\max_{\pi} \min_{D_\psi} \mathbb{E}_{s\sim\mathscr{D}}[-\log D_\psi(s, \delta(s))] + \mathbb{E}_{s\sim\pi}[-\log(1 - D_\psi(s, \delta(s)))]. \tag{20}$$

When expert actions are not accessible, we found that MConbo-GAIL can still effectively learn diverse policies without performance degradation. We attribute this to measure conditioning, which allows the algorithm to more easily infer actions from high-level state abstractions. We believe our QD-IL framework opens the door to the possibility of QD-IFO, and we look forward to future research providing more detailed studies in this area.

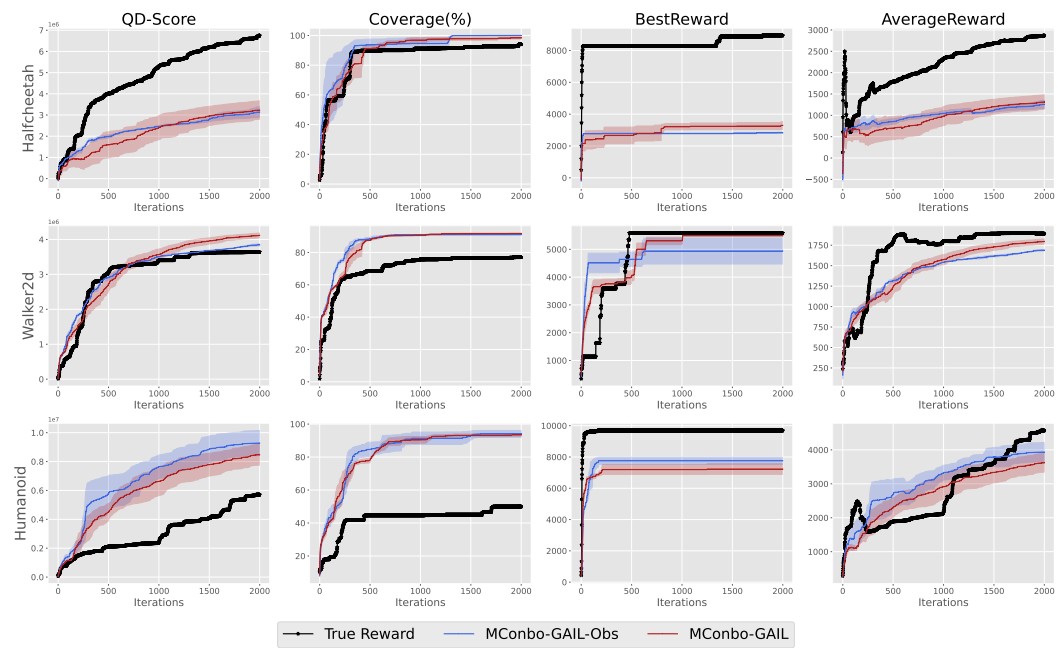

Figure 15: Comparison of MConbo-GAIL without expert action (MConbo-GAIL-Obs), with MConbo-GAIL and PPGA expert with true reward. The line represents the mean while the shaded area represents the standard deviation across three random seeds.

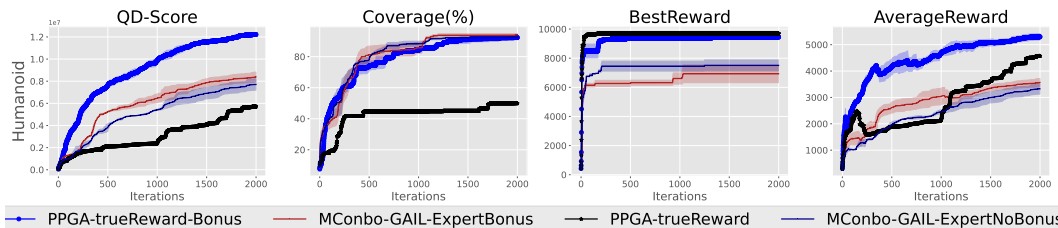

Figure 16: Apply measure bonus to PPGA with true reward: measure bonus significantly improves the PPGA with true reward. Additionally, with measure bonus, the PPGA with true reward outperforms MConbo-GAIL.

## J   MEASURE BONUS FOR QD-RL

The measure bonus is designed to promote the diversity of our QD-IL algorithm. However, we note that the measure bonus can similarly applied to QD-RL since its mechanism works regardless of the reward function it is added to. We also note that MConbo-GAIL often outperforms the expert, which gives further evidence that the true reward is not the optimal choice for QD-RL. Considering these points, we hypothesize that applying the measure bonus to PPGA with true reward can improve PPGA. We test and confirm this hypothesis on Humanoid, in which PPGA with measure bonus outperforms PPGA and MConbo-GAIL (see Figure 16).

## K   LIMITATIONS

Since the reward functions learned by GAIL and VAIL are dynamically updated, using traditional MAP-Elites to maintain the archive may not be ideal. MAP-Elites only preserves the best-performing policy at a given time, and the policy is evaluated based on the current learned reward function. Addressing these issues may further enhance the performance of our QD-IL framework.

