# OpenReview forum: "Quality Diversity Imitation Learning"
_ICLR.cc/2025/Conference — Submitted to ICLR 2025_

### Official Review · Reviewer_P8f4 · 2024-11-02

**Soundness:** 3
**Presentation:** 3
**Contribution:** 2
**Rating:** 3
**Confidence:** 3

**Summary:**

The paper proposes a novel framework called Quality Diversity Imitation Learning (QD-IL) designed to enable agents to learn a wide range of skills from limited demonstrations. The framework combines quality diversity (QD) optimization with adversarial imitation learning (AIL). It also introduces measure-based reward bonuses and measure conditioning. Experimental results show that QD-IL significantly outperforms traditional methods in Mujoco tasks.

**Strengths:**

1. The writing is clear and easy to follow.
2. The problem is well-formed and the algorithm is solid. The experimental result shows clearly the superiority of the proposed QD-IL framework.

**Weaknesses:**

1. Missing discussion of related works: there is a large literature regarding Multi-domain Imitation Learning [1] [2] [3], skill- or option-discovery [4] [5] [6] that deal with similar problems as stated in the paper. Namely, they are all designed to "find diverse solutions" so that the policies can represent diverse behaviors. If the setting is different, I suggest the authors be clearer on that; if not, these literature should be included in the discussion, and incorporated into the baselines (other than only GAIL and VAIL). It should also be suggested the advantage of the proposed QD-IL over other related works.



2. Currently, the learned diverse policies can only be read by the curves (like that in Fig.1.(a)). There lacks a formal and precise theoretical interpretation of how and why it is connected to diverse behavior patterns. How strongly does a local optimal suggest a behavior pattern and how well it is guaranteed? It would be helpful to develop more theoretical analysis in this regard.



   [1] Seyed Kamyar Seyed Ghasemipour, Shixiang Shane Gu, and Richard Zemel. Smile: Scalable meta inverse reinforcement learning through context-conditional policies. *Advances in Neural Information Processing Systems*, 32, 2019.

   [2] Lantao Yu, Tianhe Yu, Chelsea Finn, and Stefano Ermon. Meta-inverse reinforcement learning with probabilistic context variables

   [3] Jiayu Chen, Dipesh Tamboli, Tian Lan, and Vaneet Aggarwal. Multi-task Hierarchical Adversarial Inverse Reinforcement Learning. In *Proceedings of the 40th International Conference on Machine Learning*, pages 4895–4920. PMLR, July 2023.

   [4] Jesse Zhang, Karl Pertsch, Jiefan Yang, and Joseph J Lim. Minimum description length skills for accelerated reinforcement learning. In *Self-Supervision for Reinforcement Learning Workshop-ICLR*, volume 2021, 2021.

   [5] Thomas Kipf, Yujia Li, Hanjun Dai, Vinicius Zambaldi, Alvaro Sanchez-Gonzalez, Edward Grefenstette, Pushmeet Kohli, and Peter Battaglia. CompILE: Compositional Imitation Learning and Execution, May 2019.

   [6] Yiding Jiang, Evan Zheran Liu, Benjamin Eysenbach, J Zico Kolter, and Chelsea Finn. Learning Options via Compression.

**Questions:**

1. How sensitive is QD-IL to the choice of hyperparameters? It would be helpful to have a robustness/sensitivity test.
2. Can the framework be adapted to domains with high-dimensional input data, other than the three environments stated in the paper? It is also questionable to say that the "most challenging" humanoid environment, since a pixel-based input environment like Atari is clearly more challenging. It would be helpful to make it clearer.
3. Could the author give more analysis on the learned behavior patterns? Do they really correspond to "skills" in the semantic level? E.g. moving left leg, moving right leg, etc. It should be more convincing if these diversified policies can be visualized and validated as different behavior patterns for better interpretability.

---

> ### Author Response · Authors · 2024-11-22
>
> Thank you for your time and valuable suggestions. Here are our detailed responses.
> - - -
>
> **Q1**: The reviewer mentions various references regarding Multi-domain Imitation Learning and  skill- or option-discovery [4] [5] [6] and seeks to clarify how these settings are different.
>
> **A1**:  Thanks for providing these work. In multi-domain imitation learning [1,2,3], the aim is to form a **context-conditioned policy** that performs well across different tasks. Our setting is different as we are interest in **designing a large archive of policies** that represent **diverse** behaviors.
>
> In option discovery and related methods [3,4,5,6], the aim is to form low-level policies to be used within a **higher-level MDP** (e.g. an MDP with options). This line of work is related to **hierarchical RL**, where a low-level policy can be initiated under certain states and terminated upon other states (like a subgoal). In our approach, we do not seek to form policies based on their utility in a hierarchical setup; instead, we seek to form a maximally diverse and high-performing set of policies.
>
> - - -
> **Q2**: Currently, the learned diverse policies can only be read by the curves (like that in Fig.1.(a)). There lacks a formal and precise theoretical interpretation of how and why it is connected to diverse behavior patterns. How strongly does a local optimal suggest a behavior pattern and how well it is guaranteed? It would be helpful to develop more theoretical analysis in this regard.
>
> **A2**: Sorry for the confusion. Actually, Fig. 1 is merely an illustration of our motivations. Each policy corresponds to a particular behavior, and Fig. 1(a) illustrates a problem with the traditional approach of PPGA. While PPGA stores high-performing policies for each behavioral region it explores, it overemphasizes particular regions of the behavior space.  Introducing the measure bonus helps to improve this exploration process. Our approach naturally leads to high-performing solutions for diverse behaviors, thereby optimizing the objective in Definition 1. A theoretical motivation for the measure bonus can be found in Lemma 1. The main purpose is to encourage visiting new cells in the archive, thereby increasing the coverage of the behavior space.
>
> As for the learned policy archives, these can be found in Figure 6, where for each behavior in the archive we display the performance. Moreover, a directly applicable, practical evaluation method for Definition 1 is the QD Score, which we show our QD-IL approach is consistently optimizing.
>
> To improve the clarity of Fig. 1(a), we have improved the caption to illustrate the motivation that **PPGA does not sufficiently encourage exploration in itself**, and why the measure bonus can help.
>
> - - -
> **Q3**: How sensitive is QD-IL to the choice of hyperparameters? It would be helpful to have a robustness/sensitivity test.
>
> **A3**: Thanks for the suggestion. As our hyperparameter study shows (see Fig.10 in the updated paper), our results can be further improved by tuning the p and q hyperparameters related to our measure bonus. For your convenience, we hereby show the result table:
>
> | Game      | Model          | QD-Score     | Coverage | BestReward | AverageReward |
> |-----------|----------------|--------------|----------|------------|---------------|
> | humanoid  | p=0.5 q=0.5    | 7,705,305    | 92.64    | 7,504      | 3,324         |
> | humanoid  | p=0.5 q=1      | 10,563,317   | 96.44    | 8,876      | 4,383         |
> | humanoid  | p=0.5 q=2      | 10,095,293   | 95.90    | 7,313      | 4,211         |
> | humanoid  | p=1 q=0.5      | 9,933,529    | 90.72    | 7,370      | 4,358         |
> | humanoid  | p=1 q=1        | 9,959,482    | 94.04    | 7,092      | 4,237         |
> | humanoid  | p=1 q=2        | 8,719,475    | 90.62    | 5,271      | 3,847         |
> | humanoid  | p=2 q=0.5      | 9,935,292    | 93.14    | 7,879      | 4,265         |
> | humanoid  | p=2 q=1        | 8,863,207    | 95.52    | 6,638      | 3,711         |
> | humanoid  | p=2 q=2        | 9,070,480    | 95.94    | 6,669      | 3,778         |
>
> We found that p=0.5 q=1 is the most promising hyperparameter.
>
> - - -
> **Q4**: Can the framework be adapted to domains with high-dimensional input data, other than the three environments stated in the paper? It is also questionable to say that the "most challenging" humanoid environment, since a pixel-based input environment like Atari is clearly more challenging. It would be helpful to make it clearer.
>
> **A4**: Thank you for the valuable concern. We agree that there are more challenging environments. Actually, this statement was there to highlight it was most challenging within the widely-used  Mujoco environments. Since this is unclear in the abstract, the mention of most challenging has been removed.

---

> ### Author Response · Authors · 2024-11-22
>
> In the QD community, robotic locomotion tasks in MuJoCo are commonly used, such as PPGA[1], QDAD[2]. While Atari games are common in RL and IL, their implementations do not transfer to brax, which is the framework we use for hardware-accelerated simulation experiments.
>
> [1] Batra, S., Tjanaka, B., Fontaine, M. C., Petrenko, A., Nikolaidis, S., & Sukhatme, G. (2023). Proximal policy gradient arborescence for quality diversity reinforcement learning. arXiv preprint arXiv:2305.13795.
>
> [2] Grillotti, L., Faldor, M., León, B. G., & Cully, A. (2024). Quality-Diversity Actor-Critic: Learning High-Performing and Diverse Behaviors via Value and Successor Features Critics. arXiv preprint arXiv:2403.09930.
>
> - - -
> **Q5**: Could the author give more analysis on the learned behavior patterns? Do they really correspond to "skills" in the semantic level? E.g. moving left leg, moving right leg, etc. It should be more convincing if these diversified policies can be visualized and validated as different behavior patterns for better interpretability.
>
> **A5**: This is indeed a crucial point! Actually, behavior patterns can be regarded as skills or as behavioral variations of a skill, depending on one’s definition. To exemplify how a useful skill can be evolved in our archive, we note that in the Humanoid environment, our algorithm learns two distinct behavior patterns with one only using the left leg and another only using the right leg to move forward. These two can be regarded as distinct skills, particularly in terms of **dealing with changes in the environment**: if some  issue caused the left leg to be broken, then the learned “skill” enables the robot to move forward only using the right leg.
>
> To visualize the behaviors, we point to the **videos** in Supplementary Material, which demonstrate 3 different learned behaviors in each environment, chosen based on their distance to each other. For example, the Humanoid task behaviors include one moving in very small forward steps (behavior 1), one moving sideways (behavior 2), one leg shuffling (behavior 3), and one jumping behavior (behavior 4).
>
> - - -
> We hope that our responses can fully address your concerns and will be grateful for any feedback.

---

> > ### Comment · Reviewer_P8f4 · 2024-11-26
> >
> > I thank the author for the detailed feedback provided. I still have the following concerns:
> > * In Q1, the authors claim that the setting of QD-IL is to design a large archive of policies that have diverse behaviors. While I acknowledge that, I still think that learning diverse behaviors is the first step to achieving better performance in the overall task by flexibly choosing between these behaviors - in other words, for them to be conditioned on and essentially form context-based policy. So I am still concerned with what is exactly the benefit of only learning a diverse of policies, without considering how to leverage them in the most efficient way? Moreover, learning a context-based policy does not necessarily be in a multi-domain setting, it can also be applied to the current setting where the agent is supposed to finish one single task and show diverse behavior patterns. I encourage the author to make it clearer whether do you consider how to use the diverse behavior patterns or not. I take it to be yes, you still want these distinct skills to be properly leveraged in changing environments (In the current experiment of MuJoCo, for example, humanoids can use left leg - right leg separately). So,  what exactly is the benefit of QD-IL compared with the works that I mentioned regarding context-conditioned policy?
> > * In the QD-IL setting, I noticed a recent paper that hasn’t been discussed in the literature. Could the authers include the discussion for this? [1]
> >
> > 	[1] Yu, X., Wan, Z., Bossens, D. M., Lyu, Y., Guo, Q., & Tsang, I. W. (2024). Imitation from Diverse Behaviors: Wasserstein Quality Diversity Imitation Learning with Single-Step Archive Exploration. arXiv preprint arXiv:2411.06965.
> > * For the rest of the questions, I thank the authors for the clarifications and supplementary experiments.

---

> ### Author Response · Authors · 2024-11-27
>
> **Q1.** I am still concerned with what is exactly the benefit of only learning a diverse of policies, without considering how to leverage them in the most efficient way? Moreover, learning a context-based policy does not necessarily be in a multi-domain setting, it can also be applied to the current setting where the agent is supposed to finish one single task and show diverse behavior patterns.
>
> **A1.** In terms of what archives of diverse policies can be useful for, a key application of diverse policies obtained from QD algorithm is adaptation to new environments or new robot configurations. For instance, the sensors or actuators of a robot may be altered in some way, in which case performing optimisation over the archive can yield high-performing policies more quickly. This line of work has been widely studied (see e.g. [2] for one of the original studies). To give an example with Humanoid, if one leg is damaged, jumping behavior with the other leg will help the Humanoid agent to move forward.
>
> With the adaptation application above in mind, context-conditioned policies may be useful in settings where one wants to use the archive for adaptation and one can exploit a relation between the context and the optimal behavior measure. Applying the MConbo-IRL framework in such settings will be interesting for future work, which we mention in the future work discussion in the pdf.
>
> **Q2.** In the QD-IL setting, I noticed a recent paper that hasn’t been discussed in the literature. Could the authers include the discussion for this? [1]
>
> **A2.** Subsequent to our work, this paper is a recent algorithm for QD-IL, which follows our setting and was submitted to arxiv one month after our ICLR submission. While the paper follows our work closely, in terms of recognising the significance of encouraging behavior-space exploration and taking into account the measure in the design of the reward function, there are a few key differences. In particular, they introduced a measure-based bonus for single-step archive exploration, while we compute the episode-level measure bonus from the archive more straightforwardly. Moreover, they stabilized the training of adversarial imitation learning by applying Wasserstein auto-encoder (WAE) with latent adversarial training. The measure conditioning is used for learning latent variables through the reconstruction in WAE, which is different to our method that applied measure conditioning for discriminator training.  We have included the discussion of [1] in our updated paper.
>
>  [1] Yu, X., Wan, Z., Bossens, D. M., Lyu, Y., Guo, Q., & Tsang, I. W. (2024). Imitation from Diverse Behaviors: Wasserstein Quality Diversity Imitation Learning with Single-Step Archive Exploration. arXiv preprint arXiv:2411.06965.
>
> [2] Cully A., Clune J., Tarapore D. & Mouret J-B. Mouret (2015) Robots that can adapt like animals.  Nature, 521, pages 503–507.

---

### Official Review · Reviewer_vvje · 2024-11-02

**Soundness:** 3
**Presentation:** 3
**Contribution:** 3
**Rating:** 5
**Confidence:** 3

**Summary:**

This paper presents Quality Diversity Imitation Learning (QD-IL), a method that allows policies to acquire a diverse range of skills using limited demonstrations. Quality Diversity (QD) is leveraged to explore multiple solutions to optimization problems, ensuring each solution maximizes its fitness value. The authors propose a general imitation learning framework that incorporates QD within adversarial imitation learning methods, including GAIL and VAIL. The effectiveness of QD-IL is demonstrated across three simulation tasks, showcasing its potential for learning diverse behaviors.

**Strengths:**

- The paper proposes the first generic QD-IL framework that integrates QD algorithms into imitation learning.
- Introducing measure bonus into GAIL and VAIL improves policy diversity while maintaining the policy’s performance. This is demonstrated using 4 QD metrics across 3 tasks.
- The paper provides ablation studies on the measure bonus which indicate the measure bonus directly encourages the exploration of new behaviors.
- The paper is well-written with the contributions explained clearly.

**Weaknesses:**

- The measure bonus calculation requires high-level task features design, which could be challenging for long-term and complex tasks. I would assume it always needs task-specific design.
- I can’t find the details about how the authors define the behavior spaces for the 3 tasks. Maybe I missed something, but I would appreciate it if the authors could explain more about how to implement the behavior measures in the experiments.
- The experiments do not contain tasks with manipulation. It would be great to show how the QD-IL scales to other robot manipulation tasks.
- Equation 3 lacks the explanation of $\tau^{E}$.

**Questions:**

Overall, I think the idea is interesting, but I am concerned about how the method can be scaled to other complex tasks. I have the following questions:

- I don’t quite understand the Figure 1(a). Can you give a simple robot example for it? Like what is the policy space here? Does it mean a different observation space or action space?
- In Figure 1(b) Multiple behavior demos, traditional behavior cloning has the mode averaging problem, but some current methods such as diffusion-based imitation learning has the ability to learn multi-modality, right?
- QD-IL relies on the definition of behavior diverse measures $m$.  In Chapter 4.2.1, it says that the measure abstracts high-level task features. Does this mean you always need to design task-specific behavior patterns in advance? I would assume this is very hard for more complex and long-term tasks.
- The experiments do not contain any manipulation tasks. Could QD-IL be adapted to complex manipulation tasks? Could the authors give a simple case for that?
- In Figure 4, what is the x and y axis?

---

> ### Author Response · Authors · 2024-11-22
>
> Thank you for your time and valuable suggestions. Here are our detailed responses.
>
> - - -
> **Q1**: The measure bonus calculation requires high-level task features design, which could be challenging for long-term and complex tasks. I would assume it always needs task-specific design.
>
> **A1**: This point is critical. Actually, we need to clarify that the key task-specific design of complex tasks is to define the **mapping from state to measure**. After this mapping is well-defined, the calculation of measure bonus will be simple. For the mapping,
>
> - Typically, this is done based on **manual** design in terms of what the user is **interested in**. For example, in Mujoco environment, each element of single-step measure is simply designed as whether the corresponding leg touches ground (please think about it: this is actually a mapping from state to measure). This design well **suits our needs**, because we naturally hope the robot to constantly move forward without falling. In case of one leg broken, the robot can **switch** to another leg to move on.
>
> -  Due to it being a mapping from state to measure,  it is also possible to design its features **automatically**, e.g. using auto-encoders to obtain a low-dimensional representation, such as the technique in [1].
>
> Hence, the task-specific design is not problematic.
>
> [1] L. Grillotti & A, Cully (2022). Unsupervised Behaviour Discovery with Quality-Diversity Optimisation. IEEE Transactions on Evolutionary Computation.
>
> - - -
> **Q2**: I can’t find the details about how the authors define the behavior spaces for the 3 tasks. Maybe I missed something, but I would appreciate it if the authors could explain more about how to implement the behavior measures in the experiments.
>
> **A2**: Sorry for the confusion! The definition of measure (behavior space) is at the caption of Figure 2 and the description of the experiment setup (Section 5.1). In Mujoco setting, the measure function is **a vector where each dimension indicates the proportion of time a leg touches the ground**. However, the measure function can **vary** based on specific **needs** in real world application. We have improved the description of the behavior space used in the experiments.
>
> As in A2, the single-step behavior measure is implemented by a function **mapping** from state into a vector with each entry a **indicator** value (whether the corresponding leg touches ground in this state), and the episodic measure is the average of the measure of each state in this episode.
>
> - - -
> Q3: The experiments do not contain tasks with manipulation. It would be great to show how the QD-IL scales to other robot manipulation tasks.
>
> A3: This is a great extension. However, robotic locomotion tasks in MuJoCo are **commonly used** in QD community, e.g., in PPGA [1], QDAC [2]. While there are tasks such as air hockey and grasing, these are not available in standard implementations of manipulation tasks in brax, which is the framework we use for hardware-accelerated simulation experiments.
>
>
>
> [1] Batra, S., Tjanaka, B., Fontaine, M. C., Petrenko, A., Nikolaidis, S., & Sukhatme, G. (2023). Proximal policy gradient arborescence for quality diversity reinforcement learning. arXiv preprint arXiv:2305.13795.
>
> [2] Grillotti, L., Faldor, M., León, B. G., & Cully, A. (2024). Quality-Diversity Actor-Critic: Learning High-Performing and Diverse Behaviors via Value and Successor Features Critics. arXiv preprint arXiv:2403.09930.
>
> - - -
> **Q4**: Equation 3 lacks the explanation of $\tau^E$
>
> **A4**: We apologize for this undefined notation. The notation was meant to indicate state-action pairs are sampled from the demonstration set. The $\tau^E$ notation has been changed to $\mathcal{D}$ for consistency with the rest of the document.
>
> - - -
> **Q5**: The reviewer wanted to know more about the meaning of the policy space in Figure 1(a).
>
> **A5**: Sorry for confusion. Actually, figure 1(a) illustrates the fitness as a function of the policy space, i.e. the space of possible parameter settings of the policy. The figure represents the limitations of traditional policy gradient for QD optimization, namely that it will tend to explore only regions with high fitness, ignoring a large diversity of policies.
>
> The behavior space  is directly correlated to the policy space. Since it is more straightforward to give an example based on the behavior space, we now use the behavior space as the x-axis so that we can easily explain the issue and relate it to the example in Figure 2. **In short, traditional policy gradient techniques would only cover particular settings of leg contact ground times, e.g. jumping forward and walking fast, as these are highly fit, but not include more diverse behaviors that may be of interest**.
>
> - - -

---

> ### Author Response · Authors · 2024-11-22
>
> **Q6**: In Figure 1(b) Multiple behavior demos, traditional behavior cloning has the mode averaging problem, but some current methods such as diffusion-based imitation learning has the ability to learn multi-modality, right?
>
> **A6**: In Fig.1b, we pointed to the single-policy as experiencing a kind of mode averaging behavior. The reasoning is that a single policy can only represent a single point in the policy/behavior space. Therefore, designing the policy in a different manner, e.g. using a diffusion model, would not counter this issue. We kindly request that, could you please clarify the question and/or suggest some references?
>
> - - -
> **Q7**: QD-IL relies on the definition of behavior diverse measures m. In Chapter 4.2.1, it says that the measure abstracts high-level task features. Does this mean you always need to design task-specific behavior patterns in advance? I would assume this is very hard for more complex and long-term tasks.
>
> **A7**: Thank you for this valuable feedback! As we mention in A1, the behavior should be defined **in advanced** but it can be done either **manually**, by using various statistics based on the needs of the user, or using **automated** techniques, where for instance an auto-encoder could be useful. In most cases though, in quality-diversity optimization, the behavior measure is designed by hand, i.e. by choosing dimensions which the user is **interested in** from a diversity perspective. In the MuJoCo tasks, the chosen features are based on earlier works (e.g. PPGA), which uses for each leg the proportion of the time it touches the ground. Such a design leads to **useful** diverse policies, e.g. one may obtain a policy that only uses the left leg, which is useful in case the right leg is broken.
>
> In long-term tasks, one may use some **statistical properties** of the task. For Markovian Proxy Measures, the present measure we use, i.e. the proportion of touching the ground, it does not depend on the duration of the task. For long-term episodic measures, for instance, one may compute how frequently particular states have been visited, the spread of states across the episode, etc.
>
> - - -
> **Q8**: In Figure 4, what is the x and y axis?
>
> **A8**: The x axis is the proportion of time Leg 1 touches the ground and the y axis is the proportion of time Leg 2 touches the ground. Leg 1 is the left leg in Humanoid and Walker2D and it is the front leg in Halfcheetah. Leg 2 is the right leg in Humanoid and Walker2D and it is the back  leg in Halfcheetah. This description has been added to the caption of Figure 4 for improved clarity. Thanks for your concern.
>
> - - -
> Overall, we hope that our responses can fully address your concerns and will be grateful for any feedback.

---

> > ### Comment · Reviewer_vvje · 2024-11-26
> >
> > Thanks for your detailed response. From my understanding, QD-IL is then mainly designed for robotic locomotion tasks. If it can only be used in this setting, it is not very convincing. I am mainly focusing on manipulation and am not familiar with locomotion, so if I misunderstand something please let me know.
> >
> > Regarding **Q6**, I thought the multi-modality refers to the D3IL[1] and DiffusionPolicy[2] definition, which is the different solutions given a task or multi-modal action distribution.
> >
> > [1] Towards Diverse Behaviors: A Benchmark for Imitation Learning with Human Demonstrations, ICLR’24
> >
> > [2] Diffusion Policy: Visuomotor Policy Learning via Action Diffusion, RSS’23

---

> ### Author Response · Authors · 2024-11-27
>
> **Q1.** From my understanding, QD-IL is then mainly designed for robotic locomotion tasks. If it can only be used in this setting, it is not very convincing.
>
> **A1.** Robotic locomotion is a common benchmark in quality diversity (QD) optimization and quality diversity reinforcement learning (QD-RL), since there are well-implemented simulation environments to support QD learning. Our QD-IL framework is a generic method for learning diverse and high-performing policies from limited demonstrations, which is not by any means limited to robotic locomotion. The authors stress that the issue we mentioned is one of software compatibility rather than of algorithmic applicability. We are happy to apply our QD-IL framework to robotic manipulation tasks once such benchmarks are implemented in brax.  At present, the tasks of air hockey and grasping, as mentioned in one of our previous responses, are implemented in other software that is not compatible with brax.
>
> **Q2.** Regarding Q6, I thought the multi-modality refers to the D3IL[1] and DiffusionPolicy[2] definition, which is the different solutions given a task or multi-modal action distribution.
>
> **A2.** Thank you for suggesting these two papers. We can see that techniques such as diffusion policies and behavioral cloning have varying degrees of behavioral entropy according to the benchmark evaluation of D3IL, which is a dataset with demonstrations that are uniform across the behavior space.
>
> We note that the behavior entropy does not necessarily imply a set of behaviorally diverse policies, even though behavior entropy may **help to search for diverse policies**. In other words, high behavioral entropy policies are highly random in the behavior space, which implies different trials may lead to different trajectories each with different behavioral measures. This is not necessarily a desirable property as an end-goal; for instance, if each trial a different trajectory is obtained, then the policy is likely to be dangerous or unpredictable. However, we may see some role for high-entropy policies in terms of exploring the behavior space, for instance, using behavioral entropy as a bonus (very related to our measure bonus). Note that we actually considered using such a bonus but did not find a suitable implementation for our purposes.
>
> [1] Towards Diverse Behaviors: A Benchmark for Imitation Learning with Human Demonstrations, ICLR’24
>
> [2] Diffusion Policy: Visuomotor Policy Learning via Action Diffusion, RSS’23

---

> > ### Comment · Reviewer_vvje · 2024-11-30
> >
> > Thanks for your clarification. I agree that the QD-IL has the potential to extend to other tasks. However, If the QD-IL is restricted by the Brax framework, isn’t that already a big limitation? Overall, my main concern is that the experimental settings are not convincing enough. I am not very sure if it is easy to apply QD-IL to other tasks. Therefore, I will keep my score for now.

---

> > > ### Author Response · Authors · 2024-12-03
> > >
> > > **Q1.** Thanks for your clarification. I agree that the QD-IL has the potential to extend to other tasks. However, If the QD-IL is restricted by the Brax framework, isn’t that already a big limitation? Overall, my main concern is that the experimental settings are not convincing enough. I am not very sure if it is easy to apply QD-IL to other tasks.
> > >
> > > **A1.** The authors emphasise that limited software availability does not imply the algorithmic framework has limited applicability. While there are relatively many QD papers, QD optimisation is still much less explored and organised compared to RL and imitation learning. Consequently, the different benchmarks often use different software which is often incompatible. For QD-RL, this problem is even more niche, so this is why it is challenging to find suitable benchmarks compatible with our code. Since QD-RL algorithms use the Brax framework, we implemented QD-IL in this framework, which leads to only some selection of benchmarks being available.

---

### Official Review · Reviewer_wQ83 · 2024-11-04

**Soundness:** 3
**Presentation:** 3
**Contribution:** 2
**Rating:** 5
**Confidence:** 3

**Summary:**

The submission combines imitation learning and quality diversity optimization to learn an archive/set of policies that all try and maximize the unknown oracle reward function of an environment. Only expert demonstrations are provided. A modified version of Proximal Policy Gradient Arborescence (PPGA) is used as the QD-RL method to get more diverse behaviors to then generate expert demonstrations and a reference archive of expert behaviors. Measure bonus and conditioning are the two main contributions to enhance existing off the shelf inverse RL algorithms to achieve better balance of exploration and exploitation and address the challenge of localized rewards.

**Strengths:**

- The proposed modified version of GAIL, MConbo-GAIL, addresses shortcomings of GAIL such as overfitting to demonstration specific behaviors and the inability of the discriminator to generalize to new states. The changes made are sound and help encourage the discriminator to focus on high-level more generalizable features.
- The improvements shown in table 1 and figure 6 show the impact of MConbo variations of GAIL and VAIL inverse RL methods.

**Weaknesses:**

- I think an ablation on the demonstrations used to verify the robustness of the proposed method could be important. It seems that a fixed set of 4 demos per environment tested are used.
- Overall I believe this paper lacks experiments investigating the method at a deeper level. There is just one ablation and then main results, however, I think it is important to understand more about several aspects including the demonstrations and perhaps how the learned behaviors might correlate with those demonstrations. E.g. a question could be "How much of the new archive is generalization beyond demonstrations?"

**Questions:**

- Are there any training / wall clock time numbers of the proposed method? given that a gpu sim like brax is used I think wall time may be somewhat relevant to be aware of as the purpose of GPU sim is to improve training wall time.
- How well might this method fair with sub-optimal demonstrations? Or would some form of offline RL approach now be needed?

Happy to raise score to the accept range if there is more discussion on other aspects of the proposed algorithm. I primarily feel that there are a number of aspects that should be investigated to better understand the new approach with e.g., respect to its demonstration data which often influences resulting behaviors significantly. Especially given that only 4 demonstrations are used which can be more susceptible to high variance in results.

---

> ### Author Response · Authors · 2024-11-22
>
> Thank you for your time and valuable suggestions. Here are our detailed responses.
>
> - - -
>
> **Q1**: I think an ablation on the demonstrations used to verify the robustness of the proposed method could be important. It seems that a fixed set of 4 demos per environment tested are used.
>
> **A1**: That's a good point. Actually, we already performed an analysis of the number of demos in  {1,2,4} (please see Appendix H). In addition to this study, we have added an additional 10 demonstration experiment in Humanoid (see Fig.14 in the updated paper), which gives a **higher** QD score. In contrast, if the demonstration number decreases from 2 to 1, the performance significantly drops (as in Figure 13 (a) in Appendix H). These result aligns with our expectation, since our setting is actually "learning diverse behavior from limited but diverse behaviors" and it is important for the expert demonstration to be as diverse as possible, to facilitate learning diverse behaviors.
>
> In general, we can conclude that **the more diverse the demonstration, the higher the QD-score that will be obtained**. However, we primarily aims to showcase the capability of our method to learn diverse policies **from only a limited number**of demonstrations, so we only use fixed 4 demonstrations in our experiments.
>
>
> - - -
> **Q2**: The reviewer is interested in the correlation between the demonstrations and the observed behaviors for the different included algorithms.
>
> **A2**:  Thank you for your valuable concern. To compare PPGA with true reward and our proposed QD-IL algorithm in terms of the behaviors their respective archives contain, we kindly request you to point to **Figure 6**, where one can observe the diverse behaviors filling the archive with high-performing behaviors. This figure shows the behavior-performance maps, where for different 2-dimensional behavior vectors the respective elite fitness value are shown. Since there are only four demonstrations, nearly all of the policies in the archive are highly distinct from the demonstrations. The archive heatmap is **commonly used** in QD task literature to compare the learned behaviors (for example, paper [1][2][3]).
>
> In terms of interpreting the behaviors, we point to the **videos** with diverse behaviors in Supplementary Material.
>
> [1] Batra, S., Tjanaka, B., Fontaine, M. C., Petrenko, A., Nikolaidis, S., & Sukhatme, G. (2023). Proximal policy gradient arborescence for quality diversity reinforcement learning. arXiv preprint arXiv:2305.13795.
>
> [2] Fontaine, M., & Nikolaidis, S. (2023, July). Covariance matrix adaptation map-annealing. In Proceedings of the Genetic and Evolutionary Computation Conference (pp. 456-465).
>
> [3] Fontaine, M., & Nikolaidis, S. (2021). Differentiable quality diversity. Advances in Neural Information Processing Systems, 34, 10040-10052.
>
> - - -
>
> **Q3**: Are there any training / wall clock time numbers of the proposed method? Given that a gpu sim like brax is used I think wall time may be somewhat relevant to be aware of as the purpose of GPU sim is to improve training wall time.
>
> **A3**: Thanks for your valuable concern. We have recorded the wall time of each iteration in PPGA and MConbo-GAIL in humanoid environment. PPGA costs 121s and MConbo-GAIL costs 150s for each iteration. Both methods use brax to accelerate the simulation.
>
> - - -
> **Q4**: How well might this method fair with sub-optimal demonstrations? Or would some form of offline RL approach now be needed?
>
> **A4**: Sorry for the confusion. Actually, our demonstrations are already sub-optimal. Our procedure to generate demonstrations is as follows. First, we select the solutions in the archive with the top 500 highest scores. Second, from those we enforce a behavioral diversity criterion such that these demonstrations are evenly spaced in the behavior space. This leads to very diverse and therefore **sub-optimal** solutions.  So in general, the expert data are sub-optimal with regard to the global fitness optimum. Some of the solutions will be **close to local optimum** solutions.
>
> Furthermore, we think offline RL would not be possible since we do not have the reward function.

---

> > ### Comment · Reviewer_wQ83 · 2024-11-29
> > **Response**
> >
> > Thank you for clarifying however I still have some concerns. Since given the performance of the method on just one demonstration is already quite high in appendix H, it would appear this method should be tested on harder benchmarks where 1 demonstration is insufficient. Moreover regarding the interest in correlation between demonstrations and observed behaviors, while the archive filling up is good, it's hard to semantically understand as the 2D behavior vectors seem to be not interpretable. The videos are not clear as well as it sort of looks like mostly similar behaviors in one task and there aren't any videos of the demonstrations. Some direct qualitative comparisons in the supplemental or better the main paper would be great between demonstrations and learned behaviors.
> >
> > Moreover given how fast brax experiments run (just a few minutes) I hope it is not too difficult to test on the harder manipulation/locomotion tasks in brax.
> >
> > And apologies for the late review!

---

> ### Author Response · Authors · 2024-11-22
>
> **Q5**: The reviewer mentions that using only 4 demonstrations may lead to high variance in the results and that clarifying the reliability of the results would be helpful.
>
> **A5**: That point is really valuable! Actually, in our original submission, we had **already** provided the **study of number of demonstrations**. Please refer to Figure 12,13 in Appendix H, where we set the demonstration number to 4,2 and 1. In addition to these experiments, we have done further experiments where we increase the number of demos to 10, as in Figure 14 in updated paper. The result shows that the performance **significantly drops** when the number of demo decreases from 2 to 1, and the performance significantly improves when we set the number of demos to 10 (suggesting **lower variance with more demonstration**). However, even if we only have 2 but diverse demonstrations, our method can outperform PPGA with true reward, suggesting our method's ability to **learn from diverse but very limited demonstrations**.
>
> - - -
> **Q6**:. More discussion of the new approach needed for raising score.
>
> **A6**: Thanks for your suggestion. In addition to the above-mentioned experiments, we also point to further experiments which show that the **measure bonus can in principle be applied regardless of the form of reward function (whether it is true reward or learned reward)**, which means the measure bonus can **not only improve QD-IL but also improve QD-RL**. We experimentally sought to verify this on the Humanoid problem (please see PPGA-trueReward-Bonus in Fig.16 in the updated paper).
>
> For your convenience, we provided the summary of results here:
>
> | Game      | Model                      | QD-Score     | Coverage(%) | BestReward | AverageReward |
> |-----------|----------------------------|--------------|----------|------------|---------------|
> | humanoid  | PPGA-trueReward-Bonus      | 12,220,682   | 92.30    | 9,451      | 5,300         |
> | humanoid  | MConbo-GAIL-ExpertBonus    | 8,391,845    | 94.04    | 6,926      | 3,567         |
> | humanoid  | PPGA-trueReward            | 5,708,191    | 49.96    | 9,691      | 4,570         |
> | humanoid  | MConbo-GAIL-ExpertNoBonus  | 7,705,305    | 92.64    | 7,504      | 3,324         |
>
> where ''ExpertBonus'' in IL method means the demonstrations are sampled from the expert policy which is learned by PPGA  with measure bonus.
> The results confirm that the measure bonus significantly improve PPGA with true reward. Based on that, we conclude that the measure bonus can both significantly improve PPGA for QD-RL problems and improve QD-IL method.
>
> The rationales of improving these two are similar: the measure bonus helps to mitigate the local optima issue. Please note that the local optima is not such problematic when we only seek to find only **one** optimal policy (which is the traditional RL setting). However, in QD-RL setting where we aim to find multiple diverse solutions which collectively maximize the fitness, the local optima issue will be **very problematic** since we need to optimize the decision variable (policy parameter) in **each sub-region** (as illustrated in Figure (1)) but the policy parameter will intrinsically **stay around local optima point** due to the nature of gradient-based optimization method.
>
> Furthermore, in the QD-IL setting, this issue is **further exacerbated** due to the discriminator will shape the reward based on limited demonstration, so the reward will be much **more localized**. Hence, the measure bonus is designed to mitigate the problem caused by local optima, which is the **common issue** of QD-RL and QD-IL.
>
> Additionally, please note that the performance of MConbo-GAIL even exceed PPGA with true reward, which is **surprising**. We interpret this as the local optima issue mentioned before in ''PPGA with true reward'' is not solved, leading to sub-optimal performance. What verified our interpretation is that, the performance of ''PPGA with true reward'' with measure bonus outperforms MConbo-GAIL, which means the **IRL can hardly outperform RL with true reward**, when both are equipped with measure bonus.
>
> - - -
> Overall, we hope that our responses can fully address your concerns and will be grateful for any feedback.

---

> ### Author Response · Authors · 2024-12-03
>
> **Q1.** Thank you for clarifying however I still have some concerns. Since the performance of the method on just one demonstration is already quite high in appendix H, it would appear this method should be tested on harder benchmarks where 1 demonstration is insufficient.
>
> **A1.** We selected these environments as these are standard ones where our baseline, PPGA with True Reward, would perform well. As one can observe in Appendix H, Figure 13 shows a 25% improvement for 4 demonstrations vs 1 demonstration. This improvement is many times the standard deviation of the 1 demo data as can be observed from Figure 13.
>
> **Q2.** Moreover regarding the interest in correlation between demonstrations and observed behaviors, while the archive filling up is good, it's hard to semantically understand as the 2D behavior vectors seem to be not interpretable. The videos are not clear as well as it sort of looks like mostly similar behaviors in one task and there aren't any videos of the demonstrations. Some direct qualitative comparisons in the supplemental or better the main paper would be great between demonstrations and learned behaviors.
>
> **A2.** The authors have previously mentioned a comment relating to the behaviors for Reviewer P8f4, which we quote here:
> > To visualize the behaviors, we point to the videos in Supplementary Material, which demonstrate 3 different learned behaviors in each environment, chosen based on their distance to each other. For example, the Humanoid task behaviors include one moving in very small forward steps (behavior 1), one moving sideways (behavior 2), one leg shuffling (behavior 3), and one jumping behavior (behavior 4).
>
> Since we did not yet add this explanation to the pdf, we will do so in the next revision.
>
> **Q3.** Moreover given how fast brax experiments run (just a few minutes) I hope it is not too difficult to test on the harder manipulation/locomotion tasks in brax.
>
> **A3.** As one can see in the pdf, in the first paragraph of Section 5.1, it can be read that each experiment takes 2 days (assuming full availability of server). The reviewer may be used to traditional RL or imitation learning; the QD-RL and QD-IL frameworks require training a very large number of policies rather than just a single one, thereby increasing the total training time.

---

### Official Review · Reviewer_2sdj · 2024-11-04

**Soundness:** 3
**Presentation:** 2
**Contribution:** 3
**Rating:** 5
**Confidence:** 3

**Summary:**

This paper proposes a novel Quality Diversity Imitation Learning (QD-IL) framework designed to learn a diverse set of high-performing policies based on varied but limited expert demonstrations. The proposed framework, MConbo-IRL, combines elements from standard QD algorithms, such as CMA-MEGA, and adversarial imitation learning methods such as GAIL and VAIL, and introduces two key modifications: (1) adding an extra reward term that encourages exploration of policies with diverse measures (referred to as the measure bonus), and (2) adding the single-step measure of the current state as an input to the discriminator (referred to as measure conditioning). Experimental results across several continuous control tasks demonstrate that the proposed framework outperforms the baseline (PPGA trained under a single reward function learned by GAIL/VAIL) in QD metrics.

**Strengths:**

1. This paper is the first to explicitly integrate standard QD algorithms with imitation learning, though it is not the first to study imitation learning methods that learns a diverse set of policies.
* (There is another paper that studies imitation learning methods in a similar setting inspired by QD algorithms, although it diverges from standard QD algorithms: N. Justesen, M. González-Duque, D. Cabarcas, J. -B. Mouret and S. Risi, "Learning a Behavioral Repertoire from Demonstrations," 2020 IEEE Conference on Games (CoG), Osaka, Japan, 2020, pp. 383-390, doi: 10.1109/CoG47356.2020.9231897.)

2. The measure bonus, designed to encourage exploration of policies with diverse measures, is straightforward and intuitively sound.

3. The effectiveness of the designed MConbo-IRL is demonstrated through empirical results on several contibuous control tasks, which show that the MConbo- version of GAIL/VAIL overall outperforms PPGA + GAIL/VAIL and sometime outperforms PPGA + true reward in terms of QD metrics, including QD-Score, Coverage, Best Reward, and Average Reward.

**Weaknesses:**

1. The effects of measure conditioning: While the purpose of the measure bonus is straightforward, it is less clear how the measure conditioning affects the QD-IL performance. The paper states (such as in Introduction, Section 4.2.1, Appendix E) that measure conditioning (i.e., adding the single-state measure as an input to the discriminator) "enables the agent to learn broader behavior patterns". However, this claim lacks sufficient empirical support. The ablation study on measure conditioning in Appendix E shows that it impacts QD-Score and Average Reward, but the mechanism of how it impacts the policy learning remains unclear without further analysis of the algorithm's inner components.
My confusion is as follows: if adding the single-state measure input to the discriminator improves the discriminator performance, doesn't this only makes it easier for the discriminator to distinguish policy trajectories that demonstrate single-state measures absent from expert demonstrations? It seems that this could potentially discourage the policy learning to be more diverse. Could the authors clarify if this reasoning is correct, or if measure conditioning is intended to encourage diversity policy learning in a different way?

2. In Section 5.3, the observation that MConbo-GAIL achieves much higher coverage in the Humanoid environment than the PPGA expert is attributed to the "synergy between the measure bonus and CMA-MEGA". However, there are other baselines, such as PWIL and GAIL, that also achieve much higher coverage than the PPGA expert in this environment. This suggests that other factors may contribute to coverage performance in this case.

3. The expert demonstration data contains trajectories collected from multiple experts with diverse behaviors. These experts, as optimized agents from PPGA under the true reward of the environment, each optimizes their own version of the reward function. Therefore, there is no shared reward function among them. This places the PPGA + GAIL/VAIL baseline at an intrinsic disadvantage, as it is designed to learn a single reward function. It may be worthwhile to explore additional forms of baseline methods based on standard GAIL/VAIL. For example, applying PPGA + GAIL/VAIL to create a policy archive for each demonstration expert individually, then aggregating all policy archives together, could provide another baseline.

4. The paper is not very clearly written, as it sometimes references terms (often from QD algorithms) without defining them or pointing to exact external references, making the paper less self-contained.
* For instance, the format of the policy archive and the exact procedure of archive updates, are not detailed in the paper or the pseudo-code.
* Similarly, there are other parts of the pseudo-code that are not completely defined, making it difficult to understand the exact procedure.
* What is the term $p$ in Equation 2? If $p$ is a fixed constant, can it then be ignored? If that is the case, how does the definition of measure bonus in Equation 2 differ from an indicator function (in the sense that an indicator function scaled by a value $q$ is still binary)? Are $p$ and the scaling factor hyperparameter $q$ being adjusted adaptively during the training process?

**Questions:**

1. Is the concept of a measure bonus constrained to QD-IL? Is there anything that would prevent adding it to QD-RL algorithms, such as PPGA, for QD-RL problems as well?
* Currently, MConbo-GAIL can outperform the PPGA expert in the Humanoid environment, even though it is trained using demonstration trajectories from experts selected from PPGA expert's archive - which is somewhat surprising. If the measure bonus were added to the PPGA expert, could this drastically improve its performance? In that case, could MConbo-GAIL still outperform it?

---

> ### Author Response · Authors · 2024-11-22
> **Official Comment by Authors**
>
> Thank you for your time and valuable feedbacks. Here are our detailed responses.
> - - -
> **Q1**:This paper is the first to explicitly integrate standard QD algorithms with imitation learning, though it is not the first to study imitation learning methods that learns a diverse set of policies.
>
> **A1**: Thank you for your response. The paper mentioned is indeed related to our setting of imitation learning from diverse demonstrations. However, the setting of that paper has a different aim to ours, namely to formulate one policy that is a function of behavior, while we are interested in designing a behaviorally diverse archive of high-performing policies. The authors have now updated the paper to refer to this cited work and we mention how the setting of this work differs from ours.
> - - -
> **Q2**: The reviewer has some questions about measure conditioning in terms of its empirical support to demonstrate how it helps to optimize diversity.
>
> **A2**. Thanks for your valuable question. Actually, the measure conditioning is not intended to optimize diversity but rather to make the reward function more **reliable**, thus facilitating learning high-performing policies.
>
> Here is the **rationale**: we observe that the measure of one state actually serves as a **high-level abstraction** of this state. Then, the expert can demonstrate how to behave when facing some specific type of state (where the “type” means the abstraction), instead of how to behave when facing some exact state. This is helpful because the state space is quite large, so it is hard to generalize the knowledge from limited exact state-action pairs.
>
> For example, with the measure conditioning, the agent can learn to drop the right leg (action) when the left leg is not touching ground (which is reflected by measure, and also state abstraction). To improve the clarity, we have removed that sentence "enables the agent to learn broader behavior patterns”, and replaced it by “enables the agent to learn high-performing policies from limited demonstrations, improving generalization to unseen states”.
>
> - - -
> **Q3**: The reviewer mentions that there are other baselines, such as PWIL and GAIL, that also achieve much higher coverage than the PPGA expert in Humanoid, suggesting other factors affecting the coverage.
>
> **A3**: That's a really great point! In general, different reward functions will have different optimization landscapes, which may in some cases lead to higher coverage for the PPGA algorithm. Actually, we have experimented using the **zero** reward for PPGA, and the result of coverage is approximately 100% but the QD score is extremely low, which is the same as PPGA combined with other reward based on PWIL and GAIL.
>
> This phenomenon can be explained based on the inner mechanism of the PPGA algorithm. The PPGA algorithm’s objective is based on the CMA-MEGA algorithm, which is
> $g(\theta) = c_0 \cdot f(\theta) + c_1 \cdot m_1(\theta) + c_2 \cdot m_2(\theta) + \dots + c_k \cdot m_k(\theta)$, where $c_i$ are sampled from an evolutionary algorithm-based distribution. Please note that it is actually **multi-objective**, consisting of multiple terms. Optimizing one objective will potentially **inhibit** the optimization of another objective. Moreover, the first term $c_0 \cdot f(\theta)$ serves as the fitness (cumulative reward), measuring the performance of the policy, and other terms represent measures, representing the diversity of the policy. Hence, an extremely poor reward function (such as zero reward) will inhibit performance optimization but may facilitate diversity optimization. This leads to high coverage but a low QD score. On the other hand, a purely quality-based reward (such as the true reward function) will overemphasize performance and inhibit diversity optimization. This is exactly why we design the measure bonus.
>
> - - -

---

> ### Author Response · Authors · 2024-11-22
>
> **Q4**: The agent will learn a reward from each expert behavior separately and there is no shared information, which put GAIL and VAIL in intrinsic disadvantage. It may be worthwhile to explore additional forms of baseline methods based on standard GAIL/VAIL. For example, applying PPGA + GAIL/VAIL to create a policy archive for each demonstration expert individually, then aggregating all policy archives together, could provide another baseline.
>
> **A4**: This point is thought-provoking! However, our technique also **learn one single reward function**, which can be sensitive to the different experts based on the measure conditioning, and we also only use a single archive. Therefore baselines such as PPGA-VAIL/GAIL are at **no intrinsic disadvantage**.
>
> The suggested baseline would have a few disadvantages compared to our present implementation. First, the technique has limited memory scalability since for each expert one would need a separate archive. Second, separating the optimization process into different, independent optimizers limits the interchange of solutions, and it may be hard to find the genotypic hypervolume that contains the high-performing and diverse phenotypes within the same joint budget of function evaluations.
>
> - - -
> **Q5**: The reviewer shared some concerns about the writing, including more clearly defining terms, pointing to references, and clarifying aspects of the pseudo-code.
>
> **A5**: Thank you for the helpful feedback. We agree that these points could be improved. We have added more background on QD algorithms, including canonical algorithms, more details on how MAP-Elites works and how the grid-based archive is organized. We have also added the missing algorithm for updating the archive as Algorithm 2 in Appendix A.
>
> - - -
>
> Q6: The reviewer had some questions about the meaning, definition, and impact of the hyperparameters p and q in the Equation 2, and whether they are adjusted adaptively.
>
> A6: Sorry for the confusion. The hyperparameter q controls the **weight** of the measure bonus while p controls the **relative impact** of the different reward components. We use p and q simply because it is an **interpretable linear function** of the indicator function. These parameters are **fixed** throughout the optimization process. As our hyperparameter study shows (see Fig.10 in Appendix D.3 in the updated paper), our results can be further improved by tuning these hyperparameters. For your convenience, we summarize our results of hyperparameter study here:
>
> | Game      | Model          | QD-Score     | Coverage(%) | BestReward | AverageReward |
> |-----------|----------------|--------------|----------|------------|---------------|
> | humanoid  | p=0.5 q=0.5    | 7,705,305    | 92.64    | 7,504      | 3,324         |
> | humanoid  | p=0.5 q=1      | 10,563,317   | 96.44    | 8,876      | 4,383         |
> | humanoid  | p=0.5 q=2      | 10,095,293   | 95.90    | 7,313      | 4,211         |
> | humanoid  | p=1 q=0.5      | 9,933,529    | 90.72    | 7,370      | 4,358         |
> | humanoid  | p=1 q=1        | 9,959,482    | 94.04    | 7,092      | 4,237         |
> | humanoid  | p=1 q=2        | 8,719,475    | 90.62    | 5,271      | 3,847         |
> | humanoid  | p=2 q=0.5      | 9,935,292    | 93.14    | 7,879      | 4,265         |
> | humanoid  | p=2 q=1        | 8,863,207    | 95.52    | 6,638      | 3,711         |
> | humanoid  | p=2 q=2        | 9,070,480    | 95.94    | 6,669      | 3,778         |
>
> The result shows that p=0.5 and q=1 is the best hyperparameter among our choices.
>
> - - -
> **Q7**: The reviewer was wondering whether the measure bonus can be applied to QD-RL and was interested in the Humanoid problem in this regard.
>
> **A7**: This is a crucial point that we also considered during our research! Actually, the bonus can in principle **be applied to QD-RL problem**. We experimentally verify this on the Humanoid problem, and we now report results using PPGA with true reward function and measure bonus (see Fig.16 in Appendix J in the updated paper). For your convenience, we show the result table here:
>
> Model                      | QD-Score     | Coverage(%) | BestReward | AverageReward |
> |----------------------------|--------------|----------|------------|---------------|
> | PPGA-trueReward-Bonus      | 12,220,682   | 92.30    | 9,451      | 5,300         |
>  | MConbo-GAIL-ExpertBonus    | 8,391,845    | 94.04    | 6,926      | 3,567         |
>  | PPGA-trueReward            | 5,708,191    | 49.96    | 9,691      | 4,570         |
>  | MConbo-GAIL-ExpertNoBonus  | 7,705,305    | 92.64    | 7,504      | 3,324         |
>
> The measure bonus can **significantly improve PPGA for QD-RL problems**. The rationales are similar: the measure bonus helps to mitigate the local optimal issue which severely inhibits exploring diverse behavior. In the QD-IL setting, this issue is **exacerbated** due to limited demonstration. We can also see that both with measure bonus, **QD-IL can hardly outperform QD-RL**.

---

> ### Author Response · Authors · 2024-11-22
>
> Overall, we hope that our responses can fully address your concerns and will be grateful for any feedback.
>
> Please kindly let us know if anything is unclear. We truly appreciate this opportunity to improve our work and shall be most grateful for any feedback you could give to us.

---

> > ### Comment · Reviewer_2sdj · 2024-11-24
> >
> > I appreciate the authors' detailed responses. However, some of my questions and concerns remain partially or entirely unaddressed.
> >
> > * **A2**: I still have questions regarding how the proposed measure conditioning impacts the performance of QD-RL. The authors mention in the response that the rationale of adding the single-state measure to the discriminator input is based on the conjecture that the state measure can serve as a high-level abstraction of the state, and that the agent is able to better learn its policy conditioned on this high-level abstraction. However, this additional input is added to the discriminator, instead of the agent. It remains unclear whether or not this will help the agent to "behave in a specific type of state, rather than mimicking exact state-action pairs" as stated in the paper (Section 4.2.1). Even if the added input to the discriminator has impact on the discriminator's ability to distinguish a state-action pair generated by the agent from expert state-action pairs, it likely only makes the discriminator stronger, and doesn't that requires the agent to better mimic the expert's state-action pairs?
> >
> > * **A4**: My point in Weakness 3 of the review is that baselines like GAIL are algorithms designed to learn a single reward function based on expert trajectories with the assumption that the expert is optimizing for a single reward function. However, in the setting of the experiments, the expert trajectories contains multiple experts with diverse behaviors, and those experts are each optimizing for different reward functions because they are trained under different reward functions (different variations of the true reward function in PPGA). As a result, baselines like GAIL are applied in a setting where their assumption is broken and in a sense they are forced to learn a shared reward function to roughly capture the behavior of all the diverse experts, which is a disadvantage of these baselines in this setting. In this sense, the proposed method MConbo-GAIL does not have this issue as it is not really learning a single reward function because there is an added measure bonus term that is varying during training based on the changing policy archive.
> >
> > * **A6**: It seems the updated version of the paper still doesn't explain what the hyperparameter $p$ is when it appears in the definition of the measure bonus (Equation 2). I also find the use of the hyperparameter $p$ rather confusing: as a constant added to the reward function, why effect does it make in the training? Based on how $p$ is used in Equation 2 and Equation 4, It doesn't seem that it should make any difference. With the same value of $q$, the proposed method achieves different results for different values of $p$ in the hyperparameter study in Appendix D.3, which might suggest that the differences in the results are actually due to randomness in the experiments.
> >
> > * **A7**: I appreciate the authors' response to my question about applying measure bonus to QD-RL and the added discussion and experiment results about this in the appendix.
> >
> > Based on the questions and concerns that remain unaddressed, I will keep my current score.

---

> > > ### Author Response · Authors · 2024-11-24
> > >
> > > **Q1.** This additional input is added to the discriminator, instead of the agent. It remains unclear whether or not this will help the agent to "behave in a specific type of state, rather than mimicking exact state-action pairs" as stated in the paper (Section 4.2.1).
> > >
> > > **A1.** The discriminator serves as the basis for the reward, therefore with measure-conditioning the agent will get a high reward for actions that mimic the experts, and particularly so when the expert is close in state-measure, which is an abstraction of the state. Therefore, after being optimised for such a reward function, the agent will learn to mimic experts that are close in behavior. Particularly, due to it being an abstraction of the state, there is less risk of overfitting on the specific demonstration trajectories since many trajectories may map to the same state. Consequently, many behaviors can be counted as high-performing. Considering the above, the measure-conditioning will help us to achieve high quality and diversity. This description has been outlined more clearly in the pdf. With additional ablation results in Fig. 11 of Appendix E, we show that MCond outperforms GAIL in both quality and diversity.
> > >
> > > **Q2.** Even if the added input to the discriminator has impact on the discriminator's ability to distinguish a state-action pair generated by the agent from expert state-action pairs, it likely only makes the discriminator stronger, and doesn't that requires the agent to better mimic the expert's state-action pairs?
> > >
> > > **A2.** As you can also see from empirical results, the performance of our algorithm improves with the number of experts. Using measure-conditioning makes it possible to imitate diverse experts rather than a single one.
> > >
> > > **Q3.** However, in the setting of the experiments, the expert trajectories contain multiple experts with diverse behaviors, and those experts are each optimizing for different reward functions because they are trained under different reward functions (different variations of the true reward function in PPGA).
> > >
> > > **A3.** Actually, this is not the case. In our setup, the different experts that serve as the basis for the diverse demonstrations are obtained from an archive trained from PPGA. PPGA only uses a single reward function but since it is a QD algorithm, such archives contain a diverse set of policies where diversity is defined based on the behavior measure.
> > >
> > > **Q4.** As a result, baselines like GAIL are applied in a setting where their assumption is broken and in a sense they are forced to learn a shared reward function to roughly capture the behavior of all the diverse experts, which is a disadvantage of these baselines in this setting. In this sense, the proposed method MConbo-GAIL does not have this issue as it is not really learning a single reward function because there is an added measure bonus term that is varying during training based on the changing policy archive.
> > >
> > > **A4.** Yes, it can be argued, and we indeed do so, that traditional imitation learning in this setting would not work since such diverse experts would represent a different optimal behavior (and therefore implicitly an alternative reward function). The measure bonus actually does not help with this since this term is for exploration. The measure conditioning, however, helps to better imitate diverse experts rather than a single one.
> > >
> > > **Q5.** It seems the updated version of the paper still doesn't explain what the hyperparameter $p$ is when it appears in the definition of the measure bonus (Equation 2). … It doesn't seem that it should make any difference.
> > >
> > > **A5.** One can view $p$ as an additional term for staying in the episode and thereby facilitating the search for diverse behaviors. The reasoning is as follows. As the term $p$ grows, getting the maximum measure bonus in the range $[p,p+q]$  becomes less significant when compared to taking a few more steps in the episode with a lower reward of $p$. Therefore, if $p$ grows larger, the emphasis is more on staying alive, i.e. completing a full episode or going as far in the episode as possible, rather than obtaining the measure-bonus. Note that staying in the episode also makes it more likely to uncover new behaviors, therefore such an additional reward is beneficial for the search for diverse behaviors. The description has now been added clearly to the paper, and we apologise that the previous pdf did not include the description of the meaning of the parameter $p$.
> > >
> > > **Q6.**: I appreciate the authors' response to my question about applying measure bonus to QD-RL and the added discussion and experiment results about this in the appendix.
> > >
> > > **A6.** We really appreciate the suggestion and are glad that the reviewer is as pleased with results as we are.

---

> > > > ### Comment · Reviewer_2sdj · 2024-11-25
> > > >
> > > > I thank the authors for the responses to my follow-up comments:
> > > >
> > > > * Measure-conditioning: The provided ablation study in Appendix E is inadequate to show that the proposed measure conditioning is overall effective. Figure 11 contains three tasks, and MCond-GAIL outperforms GAIL on one task while the comparison is unclear for the other two tasks.
> > > >
> > > > * I believe the claim in the response that the experts achieved using PPGA are trained using a single reward function may not be accurate. While it's true that there is an original reward function in PPGA, the algorithm actually trains its diverse collection of policies under random linear combinations of the original reward function and multiple measure proxies, effectively using a variety of reward functions.
> > > >
> > > > * The additional explanation of the hyperparameter $p$ in the authors' response is helpful. I now understand that it serves as a reward term that encourages the policy to last a larger number of steps in the environment. I have a new concern that encouraging longer episodes may be introducing human knowledge of the true reward function of the test tasks like Mujoco, in the sense that going for more steps is likely positively correlated with achieving the goal of walking a longer distance in the settings. This could conflict with the goal of imitation learning, which is to learn a policy without relying on knowledge of the reward function. The choice of encouragement of longer episodes is not universal for imitation learning tasks. If tested on a different task where better performance is actually achieved by shorter episodes, for example, if the objective of a robot is to reach a target in the shortest steps possible, it seems this hyperparameter $p$ will likely have a negative impact on the IL performance due to its encouragement of longer episodes.

---

> > > > > ### Author Response · Authors · 2024-11-26
> > > > >
> > > > > The reviewer mentions a few further points of discussion, which we reply to below.
> > > > >
> > > > > **Q1.** Measure-conditioning: The provided ablation study in Appendix E is inadequate to show that the proposed measure conditioning is overall effective. Figure 11 contains three tasks, and MCond-GAIL outperforms GAIL on one task while the comparison is unclear for the other two tasks.
> > > > >
> > > > > **A1.** The figure was not clear enough so we have now additionally included a table for the results. We also give this table here for your convenience. The table shows a consistent benefit, although with varying effect size (2+ pooled std,  1 pooled std, and one very small effect). We also report the table in the updated paper.
> > > > >
> > > > > |Tasks | Model | QD-Score | Coverage | BestReward | AverageReward |
> > > > > | -------- | -------- | -------- | -------- | -------- | -------- |
> > > > > | halfcheetah | True Reward | 6,752,624 |  94.08 | 8,942 | 2,871 |
> > > > > | halfcheetah | GAIL | 2,022,500$\pm$839,063 | 67.83$\pm$16.05 | 5,115$\pm$218 | 1,167$\pm$341 |
> > > > > | halfcheetah | MConbo-GAIL | 3,235,423$\pm$750,505 | 98.32$\pm$1.21 | 3,291$\pm$430 | 1,313$\pm$291 |
> > > > > | halfcheetah | MCond-GAIL | 3,530,990$\pm$415,587 | 85.91$\pm$13.95 | 5,832$\pm$531 | 1,704$\pm$406 |
> > > > > | humanoid | True Reward | 5,708,191 | 49.96 | 9,691 | 4,570 |
> > > > > | humanoid | GAIL | 1,864,664$\pm$450,333 | 82.36$\pm$9.16 | 6,278$\pm$2,245 | 924$\pm$250 |
> > > > > | humanoid | MConbo-GAIL | 8,470,826$\pm$1,235,069 | 93.47$\pm$1.37 | 7,228$\pm$582 | 3,618$\pm$475 |
> > > > > | humanoid | MCond-GAIL | 2,194,446$\pm$316,492 | 69.07$\pm$5.54 | 7,795$\pm$397 | 1,266$\pm$105 |
> > > > > | walker2d | True Reward | 3,641,854 | 77.04 | 5,588 | 1,891 |
> > > > > | walker2d | GAIL | 2,483,228$\pm$288,096 | 69.29$\pm$4.48 | 4,031$\pm$187 | 1,429$\pm$73 |
> > > > > | walker2d | MConbo-GAIL | 4,115,586$\pm$119,161 | 91.69$\pm$0.58 | 5,491$\pm$40 | 1,796$\pm$56 |
> > > > > | walker2d | MCond-GAIL | 2,501,431$\pm$192,553 | 69.23$\pm$4.04 | 4,302$\pm$348 | 1,444$\pm$32 |
> > > > >
> > > > > **Q2.** I believe the claim in the response that the experts achieved using PPGA are trained using a single reward function may not be accurate. While it's true that there is an original reward function in PPGA, the algorithm actually trains its diverse collection of policies under random linear combinations of the original reward function and multiple measure proxies, effectively using a variety of reward functions.
> > > > >
> > > > > **A2.** Note that all of the algorithms included in the paper are using PPGA, including not only our own method Mconbo but also GAIL and other baselines. In each algorithm, we simply replaced the true reward with the imitation reward. So the PPGA algorithm is used as the base-learner for all of the QD-IL imitation learners included in the paper. Note that we did not want to prefix all algorithm names with PPGA because that makes our method in particular sound unusually long. At the same time, we also note that our own algorithm name is not very descriptive; the MConbo name by itself does not highlight the full algorithm. Therefore, the following changes were made in the paper. First, in introductory parts of Section 4 we also clearly refer to the algorithm as PPGA with MConbo; note that in the experiments, we maintain the naming convention where we do not include the PPGA prefix from all algorithms to shorten the names. Second, the PPGA-trueReward is now shortened to TrueReward. Third, in Section 5.3 in the pdf, the authors clarify that all the conditions use PPGA as the base-learner.
> > > > >
> > > > >
> > > > > **Q3.** I have a new concern that encouraging longer episodes may be introducing human knowledge of the true reward function of the test tasks like Mujoco, in the sense that going for more steps is likely positively correlated with achieving the goal of walking a longer distance in the settings. This could conflict with the goal of imitation learning, which is to learn a policy without relying on knowledge of the reward function. The choice of encouragement of longer episodes is not universal for imitation learning tasks. If tested on a different task where better performance is actually achieved by shorter episodes.
> > > > >
> > > > >
> > > > > **A3.** We note that actually a positivity bias is present in most imitation learning algorithms, as many algorithms have only positive rewards (e.g. GAIL, VAIL, GIRIL). So in this sense, we make the same assumption as is made in standard algorithms to which we compare.
> > > > > Further, it is also straightforward to remove this bias by logic (in case of domain knowledge) or hyperparameter tuning (in case of no domain knowledge). For instance, in defining the settings of $p$ for hyperparameter tuning, it is certainly possible to include settings with $p \leq 0$ in addition to settings with $p>0$, which would make it more efficient at solving such tasks as the reviewer mentions while having no particular prior knowledge of the type of task.

---

> > > > > > ### Comment · Reviewer_2sdj · 2024-11-30
> > > > > >
> > > > > > I thank the authors for the responses:
> > > > > >
> > > > > > * **A1**:
> > > > > >   * Thanks for added Table 9, but as the authors mentioned, out of the three tasks tested, measure conditioning only results in improvement of QD-Score that is beyond one standard deviation on one task. With these results, it is unclear whether measure conditioning is overall effective in improving the QD-IL performance.
> > > > > >   * Easy to fix: Currently Appendix E refers to Figure 7 for results in all three tasks, while Figure 7 only contains one task. Figure 11 seems to contain another task but is not referenced to. Similarly, the newly added Table 9 is also not referenced to by Appendix E.
> > > > > >
> > > > > > * **A2**: Section 5.3 mentions that "Each baseline learns a reward function, which is then used to train standard PPGA under identical settings for all baselines." I understand that the baseline methods all use PPGA as the QD-RL algorithm **after a reward function is explicitly learned**. My previous comment about the baselines learning a single reward function was about the stage before PPGA is applied.
> > > > > >
> > > > > > * **A3**: The authors make a good point that many baseline IL algorithms also have a bias of encouraging longer episodes due to the non-negativity of their reward function. However, I think there is still a difference between the positivity bias in the baseline IL methods and the additional positivity bias introduced in the proposed method. In the baseline IL methods, the degree of the positivity bias tends to vary with the IL quality, with state-action pairs that are more distinct from expert trajectories receiving reward values closer to 0. As a comparison, the proposed method has a constant reward $p$ at each step, which is in addition to the existing positivity bias from the discriminator reward and does not depend on the IL quality. In settings where longer episodes are indeed preferred for higher IL quality (which is likely true for the tasks used in the paper), the additional degree of positivity bias could be a factor that contributes positively in the policy performance, while the same positive effect can not be expected for IL tasks in general. This could make it more difficult to interpret some empirical results presented in the paper and compare the proposed method with the baselines.

---

> ### Author Response · Authors · 2024-12-03
>
> **Q1.**: Thanks for added Table 9, but as the authors mentioned, out of the three tasks tested, measure conditioning only results in improvement of QD-Score that is beyond one standard deviation on one task.
>
> Easy to fix: Currently Appendix E refers to Figure 7 for results in all three tasks, while Figure 7 only contains one task. Figure 11 seems to contain another task but is not referenced to. Similarly, the newly added Table 9 is also not referenced to by Appendix E.
>
> **A1.** For the first part, indeed, due to the large variance in IL it is often difficult to guarantee large effect sizes. For the second part, we will take care to update the Figure references correctly.
>
> **Q2.**: Section 5.3 mentions that "Each baseline learns a reward function, which is then used to train standard PPGA under identical settings for all baselines." I understand that the baseline methods all use PPGA as the QD-RL algorithm after a reward function is explicitly learned. My previous comment about the baselines learning a single reward function was about the stage before PPGA is applied.
>
> **A2.** All the methods are learning a single reward function. Measure-conditioning is similar to learning with multiple reward functions but in this case the “intrinsic disadvantage” that the reviewer previously mentioned is simply concluding that our method works.
>
> **Q3.**: In settings where longer episodes are indeed preferred for higher IL quality (which is likely true for the tasks used in the paper), the additional degree of positivity bias could be a factor that contributes positively in the policy performance, while the same positive effect can not be expected for IL tasks in general. This could make it more difficult to interpret some empirical results presented in the paper and compare the proposed method with the baselines.
>
> **A3.** Thanks for your comments. To test this hypothesis, we have provided the result of MConbo-GAIL with $p=0$ and the default $q=0.5$ in Humanoid. The results show that the QD-Score of MConbo-GAIL ($p=0$ $q=0.5$) is still higher than all the baselines and is approximately equal to PPGA trained with True Reward. Therefore, the results do indeed generalize to $p=0$.
>
> |Task | Model | QD-Score | Coverage | BestReward | AverageReward |
> | -------- | -------- | -------- | -------- | -------- | -------- |
> |Humanoid|True Reward | 5,708,191 |  49.96 | 9,691 |  4,570 |
> |Humanoid | MConbo-GAIL  ($p=0$ $q=0.5$) | 5,465,539$\pm$1,232,745 | 98.28$\pm$0.16 | 6,460$\pm$35 | 2,224$\pm$498 |

---

### Author Response · Authors · 2024-11-23
**We sincerely anticipate your feedback as the Discussion stage will end in 3 days.**

Dear Reviewers,

Thanks again for all your valuable comments, and we'd express our sincere apologize for our late response, due to the time cost of supplementary experiments.

We have now provided more clarifications, explanations, and experiments to address your concerns and followed the advice of all reviewers to improve our paper.

Please kindly let us know if anything is unclear. We truly appreciate this opportunity to improve our work and shall be most grateful for any feedback you could give to us.

Best regards,
Authors

---

### Meta-Review · Area_Chair_qNbr · 2024-12-18

**Metareview:**

The paper presents an imitation learning based approach for obtaining diverse skills. The reviewers have raised several concerns on the novelty of the ideas and particularly on the experimental results. I'm glad to see that the authors have tried to address these concerns. However, the reviewers still had outstanding concerns on the quality of writing, algorithmic details such as the reward function design, depth of experimental insight, limitation to locomotion / navigation tasks, and an analysis of the algorithmic design decisions on performance. I would hence recommend expanding the paper in the aforementioned directions along with incorporating the discussion with reviewers in the next version of the paper.

**Additional Comments On Reviewer Discussion:**

Overall, the reviewers were not convinced on many aspects of the evaluation of the paper and hence did not increase their score.

---

### Decision · Program_Chairs · 2025-01-22

Reject